

# Evaluation of atmospheric nitrogen inputs into marine ecosystems of the North Sea and Baltic Sea – part B: contribution by shipping and agricultural emissions

Daniel Neumann[1], Hagen Radtke[1], Matthias Karl[2], and Thomas Neumann[1]

[1]Leibniz-Institute for Baltic Sea Research Warnemünde, Seestr. 15, 18119 Rostock, Germany
[2]Institute of Coastal Research, Helmholtz-Zentrum Geesthacht, Max-Planck-Str. 1, 21502 Geesthacht, Germany

**Correspondence:** Daniel Neumann (daniel.neumann@io-warnemuende.de)

**Abstract.** Atmospheric nitrogen deposition constitutes $20\%$ to $40\%$ of the nitrogen input into the North Sea and the Baltic Sea contributing to phytoplankton growth and inducing eutrophication. Major contributors to the atmospheric nitrogen emissions in the vicinity of marine water bodies are shipping and agricultural activities. The contribution of individual emission sectors to the biogeochemical nitrogen cycle needs to be evaluated in order to assess improvement of marine water quality through emission reductions of these sectors. Hence, one focus of this modeling study was to evaluate the contribution of total, shipping-related, and agricultural-related nitrogen deposition to dissolved inorganic nitrogen (DIN), particulate organic nitrogen (PON), chlorophyll-a. A second focus of this study was to compare both water bodies with respect to the accumulation and processing of nitrogen from the mentioned sources. These two research questions were evaluated by a modeling study with the coupled physical-biogeochemical model HBM-ERGOM. The fate of atmospheric deposition in total and of atmospheric nitrogen deposition from two individual sources – shipping and agricultural activities – was traced in the marine ecosystem by a tagging method. In the Baltic Sea it was found that the atmospheric nitrogen deposition contributes up to $50\%$ to the DIN pool at individual locations during summer. On annual average, $13\%$ are contributed. Approximately $5\%$ of DIN originated from agricultural activities and $2\%$ from the shipping sector. In the western and central Baltic Sea, the shipping sector contributed nearly $5\%$. The pattern was similar for the agricultural share indicating that these two sources have a higher relevance in these regions. In the Baltic Sea, the atmospheric nitrogen shares of chlorophyll-a and bioavailable PON were $19\%$ and $18\%$, respectively, and, hence, higher than of DIN. In contrast in offshore regions only, atmospheric nitrogen shares to DIN, PON, and chlorophyll-a were on a similar level compared to each other ($20\%$ to $35\%$). This difference is caused by high DIN loads and phosphorus limitation in coastal regions of the Baltic Sea. In the North Sea, the atmospheric contribution to DIN was on a similar level but showed considerable spatial variability caused by a north-south gradient. The shipping contribution to DIN was slightly lower ($\approx 1.4\%$) and the agricultural contribution higher ($6\%$) compared to the Baltic Sea. The atmospheric contribution to chlorophyll-a and bioavailable PON was considerably lower than in the Baltic Sea and on the level of the atmospheric DIN shares, which is a result of short residence times of nutrients in the North Sea compared to the Baltic Sea. The shipping and agricultural contributions to PON and chlorophyll-a were also lower.



*Copyright statement.* TEXT

# 1 Introduction

The North Sea and the Baltic Sea are heavily impacted by anthropogenic activities (Andersen et al., 2013; Huiskes and Rozema, 1988; OSPAR, 2010; Andersen et al., 2015; Korpinen et al., 2012; Svendsen et al., 2015). Amongst other impacts, the excessive

input of nutrients from anthropogenic sources into the seawater leads to eutrophication (OSPAR, 2017a; Svendsen et al., 2015; Theobald et al., 2009). The eutrophication of North Sea and Baltic Sea has decreased in the past three decades through reduced riverine nutrient loads (Andersen et al., 2017; Svendsen et al., 2015; Gustafsson et al., 2012). But the nutrient inputs are still too high to achieve a Good Environmental Status (GES) (e.g., HELCOM, 2009; Ferreira et al., 2010; OSPAR, 2009, 2017a).

The nutrients of highest relevance are bioavailable nitrogen and phosphorus compounds. They are not only imported via

rivers but also via atmospheric deposition (OSPAR, 2017b, c; Svendsen et al., 2015; HELCOM, 2015). Atmospheric nitrogen deposition contributes approximately $20\%$ to $35\%$ to the nitrogen inputs into the Baltic Sea (HELCOM, 2013a, b; Stipa et al., 2007) and approximately $25\%$ to $40\%$ to those into the North Sea (OSPAR, 2017a, c, d). It has been reduced by approximately $30\%$ in both regions in the past two decades (OSPAR, 2017a; Svendsen et al., 2015) contributing to an improvement of the water quality. Relevant sources of atmospheric nitrogen emissions in the vicinity to marine regions are the shipping sector

(Geels et al., 2012; Matthias et al., 2010; Aulinger et al., 2016; Aksoyoglu et al., 2016; Tsyro and Berge, 1998; Jonson et al., 2015) and agricultural sector – including animal husbandry (Hendriks et al., 2016; Backes et al., 2016a; Skjøth et al., 2004; Skjøth et al., 2011; Theobald et al., 2009; Zhang et al., 2008).

Rivers and atmospheric deposition differ considerably from each other with respect to their spatio-temporal input pattern: atmospheric deposition occurs everywhere whereas rivers enter the seas only at distinct locations; oxidized nitrogen deposition

is highest in summer when nutrients are depleted in the ocean (Stipa et al., 2007; Troost et al., 2013) while rivers import fewer nutrients in this time period due to lower precipitation and less droughts. Even individual atmospheric emission sectors lead to different spatio-temporal nitrogen deposition patterns: shipping emissions take place at the sea and partly depose there. In contrast, ammonia from agricultural emissions tends to deposit close to it source and along the coast line because of its stickiness and interaction with sea salt particles (Seinfeld and Pandis, 2016; Kelly et al., 2010; Neumann et al., 2016a).

Nitrogen deposition due to residential heating is higher in winter whereas deposition due to biomass burning is higher in the other seasons. Hence, it is reasonable to consider different atmospheric emission sources individually when dealing with atmospheric nitrogen deposition.

The shipping sector is considered to contribute approximately $10\%$ and $5\%$-$8\%$ to the nitrogen deposition into the North Sea and Baltic Sea, respectively (Tsyro and Berge, 1998; OSPAR, 2017d; Große et al., 2017; HELCOM, 2009; Bartnicki et al.,

2011). The contribution to the oxidized nitrogen deposition into the Baltic Sea is higher and amounts $5\%$ to $20\%$ per year depending on the year and model (Tsyro and Berge, 1998; Bartnicki and Fagerli, 2008; Hongisto, 2014; Bartnicki et al., 2011; Stipa et al., 2007). Stipa et al. (2007) even found local contributions of up to $50\%$ in individual summer months.





From 1st January 2021 and onwards, the North Sea and Baltic Sea will be declared as $NO_X$ emission control areas (NECAs) according to MARPOL Annex VI. Oceangoing vessels, which are built after this date, have to comply with stricter $NO_X$ emission thresholds (*Tier III* thresholds). NECAs are focused on improving the air quality in coastal regions. Also the largest benefits are expected to be in this sector (Åström et al., 2018; Danish EPA, 2012). Nevertheless, lower $NO_X$ emissions lead to

lower nitrogen deposition. Because cargo vessels have a life time of approximately 25 to 30 years and because the international shipping traffic is predicted to increase (Buhaug et al., 2009; Smith et al., 2014; Danish EPA, 2012), the expected $NO_X$ emission reductions in the shipping sector are rather low in the next decade (e.g., Geels et al., 2012; Jonson et al., 2015; Hammingh et al., 2012). However, after 2030 considerable $NO_X$ emission reductions by NECA compliance and technical efficiency improvements are expected (Hammingh et al., 2012; Kalli et al., 2010; Stipa et al., 2007). Hammingh et al. (2012)

predicts shipping contribution of $6\%$ to the total nitrogen deposition without a NECA in 2030. The NECA alone would reduce the nitrogen deposition into the North Sea by approximately $2\%$ in 2030.

Agricultural emissions received increasing attention in recent two decades (e.g., EU-91/676/EEC, 1991). Large diffuse agricultural ammonia emission sources to air are located along the coast of the North Sea and Baltic Sea: the Netherlands, northern Germany, Denmark, and Poland. Particularly, animal housing and application of manure on fields are of high relevance for

ammonia emission (Backes et al., 2016b; van Pul et al., 2008; Skjøth et al., 2004; Hendriks et al., 2016). Hence, agricultural ammonia emissions closely correlate with the livestock density despite an active manure trade has been established in recent years (e.g., Hendriks et al., 2016). Ammonia deposits close to its source location or rapidly condenses on particles as ammonium – or forms these – because of its physical and chemical properties (e.g., Theobald et al., 2009; Seinfeld and Pandis, 2016, p. 31). In coastal regions, the condensation on coarse sea salt particles with short atmospheric residence time leads to increased

ammonium deposition into the sea water. This is important to note because, in contrast, nitrogen oxides have a considerably higher atmospheric residence and are, hence, transported over longer distances (e.g., Seinfeld and Pandis, 2016; Stipa et al., 2007).

Reductions of agricultural ammonia emission are expected in the next decade because of the Gothenburg Protocol (UNECE, 1999) and the EU Directive on *reduction of national emissions of certain atmospheric pollutants* (EU-2016/2284, 2016). They

force the ratifying states and EU member states, respectively, to reduce their ammonia emissions. In the North Sea region, ammonia emissions are planned to be reduced by $6\%$ to $19\%$ (OSPAR, 2017a).

In contrast to nitrogen deposition, detailed estimations on the spatio-temporal contribution of atmospheric phosphorus deposition do not exist or are on very coarse scale to the best of the knowledge of the authors (e.g., OSPAR, 2017a; Rolff et al., 2008; HELCOM, 2015, 2014; Mahowald et al., 2008). Recently, a multi-year measurement program was initiated at the Ger-

man Baltic Sea coast within HELCOM to improve the knowledge on phosphate deposition in the Baltic Sea region (HELCOM, 2017). Biomass burning, wind-blown dust from agricultural fields, and biogenic emissions (spores and parts of plants) are relevant sources for atmospheric phosphorus emissions besides desert dust (Mahowald et al., 2008). Hence phosphorus emissions are also related to rural anthropogenic activities. However, phosphorus deposition will not be in the focus of this study because we are lacking emission and, thus, deposition data.





Several studies on source apportionment of atmospheric nitrogen deposition into North Sea and Baltic have been performed in the last two decades (e.g., Bartnicki et al., 2011; Tsyro and Berge, 1998; Hongisto, 2014; Theobald et al., 2009; HELCOM, 2009; Aksoyoglu et al., 2016; Stipa et al., 2007). However, they either did not consider the fate and processing of atmospheric nitrogen in the marine environment (e.g., Bartnicki et al., 2011; Tsyro and Berge, 1998) or did not consider different source

sectors like shipping (e.g., Dulière et al., 2017; Shou et al., 2018; Troost et al., 2013; Ménesguen et al., 2018). Raudsepp et al. (2013) did both – but only for the Gulf of Finland. However, studies, which do both, are very important to evaluate the usefulness of nutrient input abatement measures and their benefits for the marine ecosystem. They are particularly valuable in describing the current ecological status and defining measures to obtain a good environmental status (GES) in the frame of the MSFD as suggested by Ferreira et al. (2010). Los et al. (2014) described such an application for testing river nutrient load

reductions.

Based on this state of knowledge and based on the preliminary work in part A of this study (Neumann et al., 2018b) we derived two research questions. To deal with these questions we used atmospheric nitrogen deposition calculated by the chemistry transport model (CTM) CMAQ (Karl et al., submitted, a; Appel et al., 2017) as input data for ecosystem model simulations with the coupled HBM-ERGOM model system (Maar et al., 2011; Brüning et al., 2014; Neumann, 2000; Neumann

et al., 2002).

  a) To what extend is atmospheric nitrogen deposition processed in biogeochemical state variables in the North Sea and Baltic Sea?

  b) What contribution do shipping- and agricultural-related nitrogen emissions have to the nitrogen deposition and to the marine nitrogen concentrations?

To (a): the share of nitrogen from deposition of the total nitrogen input is not equal to the share of nitrogen in marine biomass production. First, this is the case because the spatio-temporal nutrient release patterns of rivers and atmospheric deposition are different. Second, not only do external nutrients enter the marine system via rivers and the atmosphere but nutrients of previous years are already present. Hence, we evaluate the contribution of atmospheric nitrogen deposition to marine ecosystem parameters: dissolved inorganic nitrogen (DIN), particulate organic nitrogen (PON), and chlorophyll-a.

To (b): the shipping and agricultural emission sectors are relevant contributors to nitrogen deposition into the ocean. Hence, we considered these sectors nutrient contributions individually. This is of considerable interest in the scope of evaluating political nutrient input reduction targets and measures, such as NECAs, with respect to their effectiveness in reducing eutrophication in marine water bodies. The shipping contribution to nitrogen deposition was separately calculated from two CMAQ simulations and the agricultural fraction in the deposition was estimated from the emissions. Total, shipping, and agricultural

atmospheric nitrogen depositions were separately tagged in the HBM-ERGOM model simulations by an established method (Ménesguen et al., 2006; Neumann, 2007; Radtke et al., 2012). By using this method, nitrogen from each of the nitrogen deposition sources could be traced in the ecosystem state variables.

To the best of our knowledge, no study dealt with the contribution of different emission sectors to nitrogen deposition and the tracing of these sectors nitrogen through the marine ecosystem. Bridging this gap between atmospheric nutrient inputs





and their fate in the ecosystem is one of the major targets of this study. Atmospheric phosphate deposition is not separately considered in this study. Missing knowledge on the spatio-temporal pattern did not allow for such an assessment in the scope of this study.

## 2   Materials and Methods

### 2.1   Atmosphere

The atmospheric emissions, the chemistry transport model simulations, and the deposition data are presented in part A of this study (Neumann et al., 2018b). Therefore, they are very briefly described below.

The atmospheric chemistry simulations were performed with the Community Multiscale Air-Quality (CMAQ) model v5.0.1 (Nolte et al., 2015; Foley et al., 2010; Appel et al., 2017). CMAQ calculates atmospheric concentration (gas phase and particle phase), wet deposition, and dry deposition of air pollutants. The cb05tump mechanism calculated the gas phase chemistry (Sarwar et al., 2007; Whitten et al., 2010; Yarwood et al., 2005) and the aero5 mechanism, which is based on ISORROPIA v1.7, the aerosol chemistry (Fountoukis and Nenes, 2007; Sarwar et al., 2011). For the dry deposition parameterization was presented by Binkowski and Shankar (1995) and Binkowski and Roselle (2003) and the wet deposition parameterization by Foley et al. (2010).

The meteorological forcing data were extracted from the coastDat3 database (HZG, 2017) and were processed by a modified version of CMAQ's Meteorology-Chemistry Interface Processor (MCIP) (Otte and Pleim, 2010).

The land-based emissions were calculated by SMOKE for Europe (Bieser et al., 2011) with hourly resolution. The included agricultural emissions are subject to high uncertainty. The reasons are as follows. (a) The agricultural ammonia emissions strongly depend on time of the application of manure. It is known in which periods of the year and after/before which dates manure is applied. However, no information on the exact days of application per field is available. (b) Improved ammonia emission time profiles for SMOKE for Europe were derived by Backes et al. (2016a). However, older less precise ones of Bieser et al. (2011) were used these study's emissions due to technical issues. (c) Soil and crop types were not known in sufficient spatial resolution in most regions or they were not known at all. Additionally, SMOKE emission factors have been derived for North American soils but no bijective mapping exists between US American and European soil type categorizations.

The shipping emissions were created as described in Jalkanen et al. (2012) for the Baltic Sea based on data of the automatic identification system (AIS). The method by Jalkanen et al. (2012) is considered to by state-of-the-art. Sea salt emissions were calculated online (Gong, 2003; Kelly et al., 2010; Neumann et al., 2016b) and other natural marine emissions were not included.

Two emission data sets – one with all emission sources and another without shipping emissions – were prepared for the CTM simulations. The shipping contribution to the nitrogen deposition was taken as difference between both model runs. The agricultural nitrogen contribution to the nitrogen deposition was estimated to be $95\,\%$ of the total $NH_3 + NH_4^+$ deposition. This is a realistic assumption because $95\,\%$ to $100\,\%$ of the ammonia emissions originate from agricultural activities (Bieser, 2018; CEIP, 2017). Emissions of oxidized nitrogen compounds and emissions of wind-blown dust are not considered. Therefore, we underestimate the agricultural nitrogen contribution in this study. The uncertainty in the agricultural emissions themselves





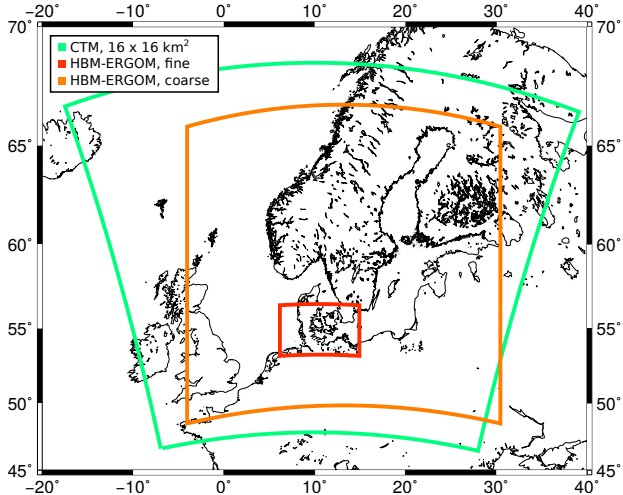

**Figure 1.** Model domains. The extent of the inner model domain of the chemistry transport model (CTM) is plotted in green. The extents of the HBM-ERGOM model domains of $5' \times 3'$ (coarse) and $50'' \times 30''$ grid cell size (fine) are represented by the orange and red frames, respectively. Figure taken from Neumann et al. (2018b).

and, additionally, in the estimation of the agricultural fraction from the cumulated emission results in a high uncertainty in agricultural nitrogen deposition, which is difficult to quantify.

Because the creation of emission data sets requires a lot of hand word and is very labor intensive, only a one-year emission data set was provided.

5 The simulations were performed on a grid of $16 \times 16 \text{ km}^2$ horizontal resolution with 30 vertical z-layers. It covers the North Sea, the Baltic Sea and the adjacent land masses (Fig. 1) and was one-way nested into a coarser resolved grid. Two simulations – one with all emission sources and another without shipping emissions – were performed for the year 2012 and hourly output data were written. The spin-up period was 30 days.

Daily mean oxidized and reduced nitrogen deposition fields were calculated from the hourly wet and dry deposition output 10 of these compounds:

- Oxidized nitrogen: NO, $NO_2$, $HNO_3$, $N_2O_5$, $NO_3^-$, $NO_3$, PAN (peroxyacetyl nitrate), HONO, PNA (peroxynitric acid; only wet deposition)

- Reduced nitrogen: $NH_3$, $NH_4^+$

The fields were bilinearly interpolated onto the HBM-ERGOM model grids as model input.




## 2.2 Ocean

The simulations in this study were performed with the coupled physical biogeochemical model HBM-ERGOM. As for the atmospheric model, the oceanic part has been also presented in part A (Neumann et al., 2018b) and will be only briefly described here.

The ocean physics were modeled by the HIROMB-BOOS-Model (HBM) (Brüning et al., 2014; Poulsen et al., 2015; Dick and Kleine, 2007). HBM has been extensively validated for Copernicus Marine Environment Monitoring Service (CMEMS) and by Wan et al. (2012) and Brüning et al. (2014).

The Ecological ReGional Ocean Model (ERGOM) (Neumann, 2000; Neumann et al., 2002) has been developed and used for ecosystem model studies in the Baltic Sea (e.g., Kuznetsov et al., 2008; Lessin et al., 2014; Miladinova and Stips, 2010; Neumann et al., 2015; Radtke et al., 2012; Neumann and Schernewski, 2005; Schernewski and Neumann, 2005).

The coupled HBM-ERGOM model system has been first applied Maar et al. (2011) and is intended to cover North Sea and Baltic Sea ecosystem features. This study's version is presented in the first part of this study (Neumann et al., 2018b) and in the Supplement. Nitrogen from the sources of interest was tagged in the ERGOM simulations (see further below for the specific tagged sources). The method used for this purpose was described by Ménesguen et al. (2006) and implement in ERGOM by Neumann (2007) and Radtke et al. (2012). It is denoted as TBNT (trans-boundary nutrient transport) method in some publications. It is further described by examples in Neumann et al. (2018b).

The North Sea and Baltic Sea are covered by two two-way nested model domains of different resolution, denoted as fine (Fig. 1, red) and coarse (Fig. 1, orange) grid domain:

- **coarse:** regular lat-lon grid of $5' \times 3'$ (Longitude $\times$ Latitude) horizontal resolution, 36 vertical layers in z-star coordinates,

- **fine:** regular lat-lon grid of $50'' \times 30''$ horizontal resolution, 25 vertical layers in z-star coordinates.

Initial conditions for HBM were generated from a regular model run of the BSH. ERGOM was spin up for two months (Nov. + Dec. 2011). No further spin-up period for the model was necessary except for the tracer tagging (see below). River data – runoff and nutrient loads – and data on the exchange with the Atlantic Ocean were taken from the BSH default setup (Brüning et al., 2014; Janssen et al., 1999; Maar et al., 2011). The meteorological conditions at the sea surface are taken from operational weather forecasts of the German Weather Service (DWD). The atmospheric nitrogen deposition data were used as presented in Sect. 2.1. The atmospheric phosphorus deposition was set to $0.471$ µmol m$^{-2}$ d$^{-1}$ ($\approx 5.33$ mg m$^{-2}$ a$^{-1}$).

The source attribution method is applied to track the whole nitrogen from atmospheric deposition and the nitrogen from agricultural- and shipping-related atmospheric deposition. The calculation of the nitrogen deposition of the shipping and agricultural sectors to the total nitrogen deposition is described in Sect 2.1. In order to improve the readability of the text, the sources of tagged tracers are written with the source's name as subscript: *"nitrogen from atmospheric deposition in nitrate"* is written as nitrate$_{atmos}$ and *"nitrogen from shipping emissions in dissolved inorganic nitrogen"* is written as DIN$_{ship}$.

The biogeochemical model was run for five years with tagged tracers. The first four years were considered as spin-up and only the fifth year was evaluated (Neumann et al., 2018b; Los et al., 2014). The same forcing – rivers, atmosphere, and Atlantic





**Table 1.** List of stations considered for parts of the evaluation. ICES abbreviates the Data portal of the International Council for the Exploration of the Sea (ICES), DOD abbreviates the database of the Deutschen Ozeanographischen Datenzentrum (DOD, German Oceanographic Data Center), and IOWDB abbreviates the Oceanographic Database of Leibniz Institute for Baltic Sea Research Warnemünde.

| Station Name | Lat | Lon | Data Origin |
|---|---|---|---|
| Anglia Anopensea Wa | 51.98 | 2.07 | ICES |
| Terslg235 | 55.17 | 3.16 | ICES |
| P8 IV | 54.15 | 7.57 | DOD |
| OMBMPK8 | 54.72 | 12.78 | IOWDB |
| BY15 | 57.33 | 20.05 | IOWDB |

Ocean – was used for each year and the physical model was restarted from its initial conditions each year. This allowed the tagged tracers to converge to a steady-state. Inter-annual variations are only due to the biogeochemistry and not caused by changes in the model forcing or by the ocean physics. The silicate concentrations would be depleted in some parts of the Baltic Sea after three to four years due to underestimated sources or recycling (see for details Neumann et al. (2018b)). Hence, the
silicate tracer concentrations were restarted from the initial conditions each year.

### 2.3   Presentation of results

The vertically averaged model concentrations of dissolved inorganic nitrogen (DIN), bioavailable particulate organic nitrogen (PON), and chlorophyll-a of the upper five model layers are considered and plotted for the evaluation of the nutrient tagging studies. The depth of these layers is variable – because of the vertical z-star coordinates used – but approximately corresponds
to the top $12$ m of the water column. In this study, PON describes only the bioavailable particulate nitrogen concentration. Some of the evaluation is performed at five stations, four of which have been considered in part A of this study (Neumann et al., 2018b) for the model validation. The stations' locations are listed in Table 1 and marked in the first figure of Sect. 3.2 (Fig. 4). Measurement data are plotted in the latter figure. The origin of that data is listed in Table 1.

A validation of the biogeochemical model simulations has been presented in part A of this study (Neumann et al., 2018b)
and is not duplicated in this part B. The station Terslg235 was not considered in the validation because of low data availability. Instead, the station BOOMKDP ($53.38°$ N, $5.17°$ E) was considered for validation in part A.

### 3   Results and Discussion

The results section is structured in three subsections. The Sect. 3.1 deals with the atmospheric nitrogen deposition and briefly describes the contribution from the shipping and agricultural sector. Section 3.2 contains the actually interesting results. In
Sect. 3.3 a comparison with other similar studies in presented.



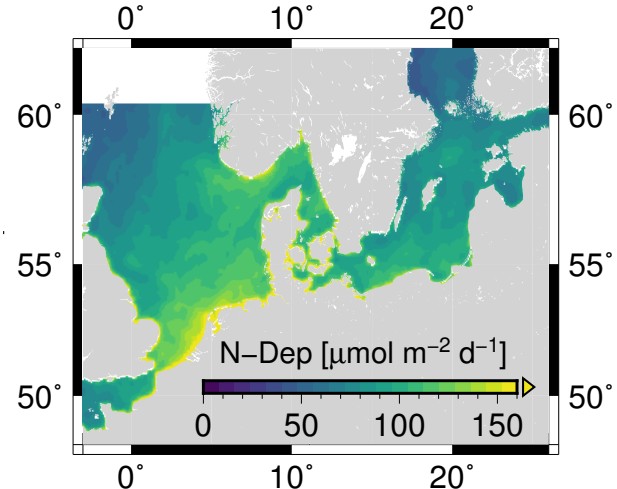

**Figure 2.** Annual average nitrogen deposition. Figure taken from Neumann et al. (2018b).

### 3.1 Atmospheric Deposition

A general overview of the nitrogen deposition used and a brief comparison with EMEP data were presented in part A of this study (Neumann et al., 2018b). A more detailed evaluation of the nitrogen deposition data of the Baltic Sea is provided in Karl et al. (submitted, a). Figure 2 gives an overview of the total annual average nitrogen deposition. The three panels in Fig. 3 show the shipping-related, agricultural (including animal husbandry), and residual nitrogen deposition from left to right. The residual nitrogen originates from long-range-transport and from anthropogenic and natural land-based non-agricultural emissions – such as road transport, power plants, and industrial facilities. Table 2 hold corresponding numbers.

The nitrogen deposition is highest along the coastline and shows steep gradients there. These gradients are caused by the dry deposition of coarse particles and of sticky gaseous substances – particularly ammonia – from land-based sources. The dry deposition is enhanced by interaction between gaseous nitrogen species – nitric acid formed from $NO_X$ and ammonia of primary emissions – and coarse sea salt particles (Neumann et al., 2016a). The patchy patterns in the open sea primarily are the result of nitrogen wet deposition.

Deposition in coastline grid cells might also artificially be increased. This has been mentioned in part A of this study already and is quoted here: *"Dry deposition over land is higher than over sea. The nitrogen deposition to the ocean is calculated during the post-processing of model data via the land-sea-fraction for each grid cell. However, the enhanced deposition over land is not considered in this process. Thus, the sea-fraction of the nitrogen deposition is overestimated and the land-fraction is underestimated. Moreover, the grid resolution of the CMAQ simulations was coarser than that of the HBM-ERGOM simulations adding another source for possible under- or overestimation – depending on the region. Neumann et al. (2018a) evaluates these issues for the western Baltic Sea by comparing two differently resolved nitrogen deposition data sets."* (Neumann et al., 2018b).





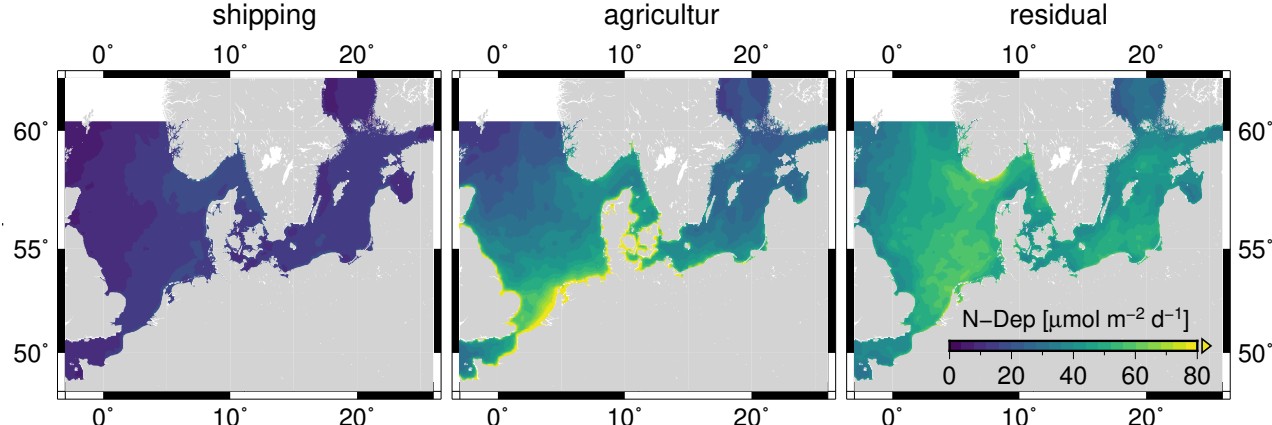

**Figure 3.** Annual average nitrogen deposition from shipping emissions (left), agricultural ammonia emissions (center), and residual emission sectors (right).

The shipping-related, agricultural (including animal husbandry), and residual nitrogen deposition are plotted in Fig. 3. The absolute and relative contributions of these source sectors are given in Table 2.

The shipping-related nitrogen deposition does not peak along the major shipping routes but it is widely distributed and shows a blurry pattern. Shipping emissions consist of species with low dry deposition propensity. Therefore, air pollutants from ships are carried over longer distances and are not strongly deposited in the vicinity of their source. Regions of maximum nitrogen deposition are around Denmark and at some spots in the Baltic Sea. Generally, the shipping contribution is lower than the agricultural contribution and the residual contribution. Although the shipping emission data set is assumed to be of high quality as described in Sect. 2.1, the atmospheric processing and deposition of shipping-$NO_X$ is impacted by immissions of other emission sectors. Hence, issues with sea salt, agricultural ammonia, or road traffic emissions impact the shipping-related nitrogen deposition and add uncertainty to it. To assess the quality of the shipping-related nitrogen deposition it is compared to literature values.

Expressed in numbers, the shipping sector has a contribution to the nitrogen deposition of $13\%$ in the North Sea and of $16\%$ in the Baltic Sea. For the Baltic Sea, these values are relatively high compared to other literature values. Bartnicki et al. (2011) found the relative contributions of approximately $8\%$. Other studies estimated a similar or lower relative contribution (HELCOM, 2009, 2013b). Also the absolute contributions were found to be lower in other studies with 16 to 18 kt N a$^{-1}$ in Bartnicki et al. (2011), 15 kt N a$^{-1}$ in Hongisto (2014), and approximately 22 kt N a$^{-1}$ in Karl et al. (submitted, a).

For the North Sea, the relative contribution is closer to other literature values. Tsyro and Berge (1998) found the shipping sector to contribute $5\%$ to $15\%$ of the nitrogen deposition into the North Sea in the 1990s. According to OSPAR (2017d), the North Sea shipping sector contributed approximately $9.3\%$ to the North Sea nitrogen deposition in 2012 (44 kt N a$^{-1}$), whereas the combined North Sea and North-East Atlantic shipping sector contributed $12.3\%$ (58 kt N a$^{-1}$). As for the total nitrogen deposition, OSPAR (2017d) estimated a higher absolute shipping-related nitrogen deposition than used in this study.



**Table 2.** Absolute (in $\mathrm{kt\ N\ a^{-1}}$) and relative (in %) contributions of the emission source sectors shipping and agriculture to the nitrogen deposition into the North Sea and Baltic Sea.

| *nitrogen* | North Sea | | Baltic Sea | | Both Seas | |
|---|---|---|---|---|---|---|
| *deposition* | absolute | relative | absolute | relative | absolute | relative |
| | $\left[\mathrm{kt\ N\ a^{-1}}\right]$ | [%] | $\left[\mathrm{kt\ N\ a^{-1}}\right]$ | [%] | $\left[\mathrm{kt\ N\ a^{-1}}\right]$ | [%] |
| shipping | 38 | 13 | 29 | 16 | 67 | 14 |
| agriculture | 115 | 40 | 70 | 37 | 185 | 39 |
| **total** | 290 | 100 | 189 | 100 | 479 | 100 |

Agricultural emissions are subject to high uncertainty as documented in Sect. 2.1. Unfortunately, no published data sets on the agriculturall related nitrogen deposition were available for a comparison and validation. In this study, agricultural emissions consist of ammonia only. Thus, the agricultural contribution to nitrogen deposition is underestimated. Nevertheless, the agricultural contribution to the nitrogen deposition is very high in the southern North Sea and southern Baltic Sea along

the coast. Ammonia is very sticky. It does not reside in the gas phase for a long time. Instead, either it condenses on particles as ammonium or it forms new particles or it deposits dry to the ground. In the case of condensation on coarse particles, the ammonium is rapidly removed by as dry deposition. Hence, agricultural-related nitrogen deposition peaks in the vicinity of agriculturally used areas and, particularly, downwind to these. Therefore, it is (a) higher at near-shore locations than at the open-ocean locations and (b) higher eastward of Denmark – downwind – than westward of Denmark. Long-range transport

and removal by precipitation are of low relevance.

In contrast to the agricultural nitrogen deposition, the residual nitrogen deposition, which is mainly oxidized nitrogen, is dominated by wet deposition processes and by dry deposition of nitrogen species with long atmospheric residence time. Hence, the spatial patterns of these two sources look quite different although they are in the same order of magnitude.

### 3.2 Contribution of different sectors

The contribution of total, shipping-related, and agricultural-related nitrogen deposition to the marine nitrogen concentrations is considered in this section. DIN, bioavailable PON (PON), and chlorophyll-a concentrations in the surface water at five stations and for the whole North Sea and Baltic Sea are evaluated (Fig. 4; Table 3). Four of them have been considered in the validation section (OMBMKP2, BY15, P8 IV, and Anglia Anopensea Wa). One new location in the Central North Sea is regarded (Terslg235). Additionally, the spatial distribution of the annual mean contribution of these sources to DIN, PON, and

chlorophyll-a are plotted further below (Fig. 6).

At the two Baltic Sea stations (Fig. 4), the atmospheric nitrogen deposition approximately contributes 29% to 30% to the DIN pool on annual average (Table 3). The $\mathrm{DIN_{agri}}$ is approximately 1/3 of $\mathrm{DIN_{atmos}}$ and makes up 10% to 11% of total DIN on annual average. The $\mathrm{DIN_{ship}}$ is low with 4% to 5%. The contributions at these stations are approximately twice as high





**Table 3.** Relative contribution in percent of nitrogen from atmospheric sources to DIN (row 1-3), PON (row 4-6), and chlorophyll-a (row 7-9). Data at five stations and for the whole North Sea and Baltic Sea is shown (different columns). The stations left of the North Sea column are located in the North Sea and the stations between the North Sea and Baltic Sea columns are located in the Baltic Sea.

| [%] | | Anglia Anopensea Wa | Terslg235 | P8 IV | **NorthSea** | OMBMPK8 | BY15 | **BalticSea** |
|---|---|---|---|---|---|---|---|---|
| DIN | total atmosphere | 10.6 | 7.0 | 10.4 | 12.5 | 28.5 | 29.5 | 12.6 |
| DIN | shipping | 1.0 | 0.9 | 1.1 | 1.4 | 3.8 | 4.6 | 1.9 |
| DIN | agriculture | 5.1 | 2.3 | 5.1 | 6.3 | 11.5 | 9.6 | 4.8 |
| PON | total atmosphere | 10.6 | 7.2 | 10.2 | 10.4 | 27.2 | 28.9 | 17.6 |
| PON | shipping | 1.1 | 1.0 | 1.1 | 1.3 | 3.9 | 4.9 | 2.7 |
| PON | agriculture | 5.0 | 2.5 | 5.1 | 4.5 | 11.2 | 9.7 | 6.6 |
| Chl-a | total atmosphere | 11.3 | 6.5 | 10.1 | 10.8 | 28.1 | 29.9 | 19.5 |
| Chl-a | shipping | 1.0 | 0.9 | 1.0 | 1.3 | 3.9 | 5.1 | 3.1 |
| Chl-a | agriculture | 5.3 | 2.1 | 4.9 | 4.5 | 11.4 | 10.0 | 7.0 |

compared to the Baltic Sea average. This difference is reasonable because the atmospheric nitrogen contribution is highest in the western Baltic Sea and in the eastern Gotland Basin (Fig. 6).

DIN accumulates during winter and the $DIN_{atmos}$ share ($DIN_{agri}$ and $DIN_{ship}$ as well) takes quite constant values. When the phytoplankton starts growing and DIN is consumed in spring, the relevance of the atmospheric DIN increases. The atmo-
spheric DIN shares remain high during summer but decrease considerably in autumn. This inverse peak is caused by cyanobacteria as described further below. In late autumn the atmospheric DIN shares rise again to similar values as in January.

The $DIN_{agri}$ share is lower at BY15 than at OMBMPK8. This is due to less agricultural activity upwind to the central Baltic Sea compared to the western Baltic Sea. In contrast, the $DIN_{ship}$ share is similar at both stations. Thus, the shipping sector contributes more to the atmospheric nitrogen deposition at Gotland.

The annual average atmospheric contributions to the chlorophyll-a concentrations (Table 3) at the two stations are similar to their shares of DIN. They are higher at these stations than on Baltic Sea average – as for DIN. However, the chlorophyll-a Baltic Sea average contributions are approximately $50\%$ higher than those for DIN (Table 3).

The atmospheric Chl-a shares show no clear trend at OMBMPK8 until late summer. In contrast at BY15, the shares rise considerably until nearly $50\%$ in summer. This indicates that atmospheric nitrogen deposition is very important for the nutri-
ent supply in this time period. The total chlorophyll-a concentrations peak in August (OMBMPK8) and September (BY15), whereas the $Chl\text{-}a_{atmos}$, $Chl\text{-}a_{agri}$, and $Chl\text{-}a_{ship}$ concentrations show no signal. Instead, their shares decrease considerably. Cyanobacteria bloom in this period. The cyanobacteria fix atmospheric $N_2$, which is not tagged, and, as a result, the $Chl\text{-}a_{atmos}$, $Chl\text{-}a_{agri}$, and $Chl\text{-}a_{ship}$ concentrations remain unchanged (Fig. 4, bottom right). In late autumn these shares rise again – partly due to a flagellate bloom. Notably, the relative atmospheric contribution to chlorophyll-a is slightly lower than
its relative contribution to DIN.





**Figure 4.** Nitrogen in DIN (1st and 2nd row) and chlorophyll-a (3rd and 4th row) from different sources. In the rows 1 and 3, the total concentrations and the contributions from total, agricultural-related, and shipping-related atmospheric deposition are plotted (see legend top left) at five locations (see legend top right). Symbols indicate measurements and the lines indicate model data. In the rows 2 and 4, the relative contribution of the three types of atmospheric nitrogen to the DIN and chlorophyll-a concentrations is show. The order of the time series plots equals the order of the stations on the map from the left to the right. The color of the symbols in the time series plots equals the colors of the respective stations. DIN measurements at Anglia Anopensea Wa are not available for the particular year.

The spatial pattern of the total PON and chlorophyll-a concentrations look quite different (Fig. 5). However, the relative shares of the shipping, agricultural, and total atmospheric nitrogen are similar in PON and chlorophyll-a in the Baltic Sea. Hence, PON is not described here.





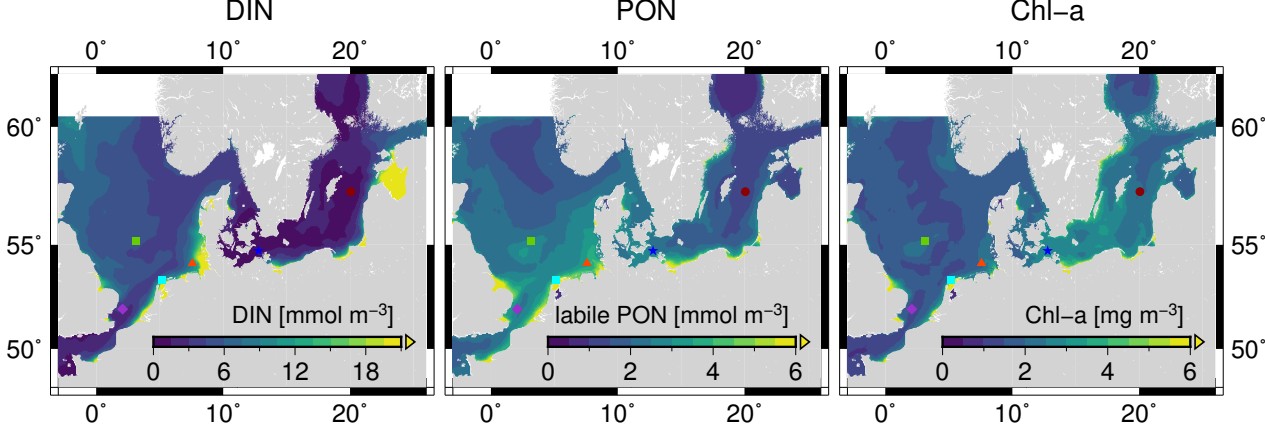

**Figure 5.** Annual mean concentrations of DIN, PON, and chlorophyll-a concentrations (left to right) of the year 2012 after four years of spin-up. The six colored symbols indicate the locations of the measurement stations in Fig. 4 and Table 3 (left to right: Anglia Anopensea Wa, Terslg235, BOOMKDP, P8 IV, OMBMPK8, and BY15).

The atmospheric nitrogen deposition has a lower relevance at the North Sea stations. At the two stations in the vicinity to land (P8 IV and Anglia Anopensea Wa) the $DIN_{atmos}$ share makes up $8\%$ to $10\%$ on annual average, of which approximately 1/2 is $DIN_{agri}$ and 1/10 is $DIN_{ship}$ (Table 3). The inter-annual pattern differs at both stations. On North Sea average, the DIN shares are slightly higher. This is reasonable because the two stations are considerably impacted by nutrient loads from rivers.

The impact of atmospheric nitrogen emissions from land reaches further into the ocean than the impact of rivers. Hence, the contribution of atmospheric deposition is lower in the sphere of influence of large rivers as the Elbe Rhine (Fig. 6). Nevertheless, the stations' DIN shares are closer to the North Sea average DIN share than it is the case for the Baltic Sea stations and average.

At Anglia Anopensea Wa the $DIN_{atmos}$ share decreases in spring and peaks in summer. The decrease correlates with a peak in the total DIN concentrations. The latter peak probably originates from riverine or Atlantic DIN, which is untagged. The

DIN is consumed by phytoplankton and transport further east in the summer months making atmospheric nitrogen deposition an important source of external DIN, which leads to the higher $DIN_{atmos}$ shares. In autumn and winter however, more DIN is transported into this region from the English Channel. This DIN is supplied from the Northeast Atlantic Ocean and is untagged.

In contrast at P8 IV, $DIN_{atmos}$ share decreases rapidly in spring and, then, steadily rises until December. From spring to autumn large amounts of external DIN are supplied from German rivers. But in contrast to Anglia Anopensea Wa, the water

masses arriving in the German Bight from the west are already enriched in nitrogen of atmospheric origin. Therefore, the relative contribution of atmospheric deposition rises throughout the year.

The station Terslg235 is more distant to the coast. The absolute DIN concentration is on a similar level as at Anglia Anopensea Wa but there is less input of external DIN: land-based sources of atmospheric emissions are far distant. Therefore, the shares of $DIN_{atmos}$ and $DIN_{agri}$ are lower than at the other two North Sea stations. In contrast, the share of $DIN_{ship}$

is only slightly lower than at the other stations (Fig. 4 pink line and Table 3).





**Figure 6.** Annual mean relative contribution of total, shipping-related, and agricultural-related nitrogen deposition (top to bottom) to the DIN, PON, and chlorophyll-a concentrations (left to right) in of the year 2012 after four years of spin-up. The colored scaling within each row is equal but it differs between different rows. The six colored symbols indicate the locations of the measurement stations in Fig. 4 and Table 3 (left to right: Anglia Anopensea Wa, Terslg235, BOOMKDP, P8 IV, OMBMPK8, and BY15).

The relative contribution of the atmospheric deposition to chlorophyll-a – total, agriculture, and shipping – takes very similar values than to DIN. Because in ERGOM no cyanobacteria or other $N_2$-fixing species grow in the North Sea, no external untagged nitrogen is introduced into the water from the atmosphere. Thus, Chl-a$_{atmos}$, Chl-a$_{agri}$, and Chl-a$_{ship}$ shares do not decrease in late summer and autumn as in the Baltic Sea.



### 3.2.1 Comparing North Sea and Baltic Sea

The absolute $DIN_{atmos}$, $DIN_{agri}$, and $DIN_{ship}$ concentrations are quite similar at P8 IV and at the two Baltic Sea stations (Fig. 4, top row). Their relative shares are lower at P8 IV (Fig. 4, second row) because larger amounts of DIN are introduced into the German Bight by rivers – particularly by the Elbe river. This is a general difference between the North Sea and Baltic

Sea as Fig. 5 visualizes: The DIN and PON concentrations are higher at the North Sea stations. This is consistent with the results of Maar et al. (2011). The nitrogen deposition per square meter is of similar magnitude in both ocean areas and, thus, the $DIN_{atmos}$ and $PON_{atmos}$ shares (the shipping and agricultural shares as well) are higher in the Baltic Sea. Additionally, nutrients remain in the Baltic Sea for longer time periods than in the North Sea (Sect. 3.3 in part A, Neumann et al. (2018b)), which increases the relative contribution of one tagged source.

The DIN and PON concentrations are very high in German Bight as Fig. 5 illustrates compared to other North Sea regions and the Baltic Sea. This is partly caused by the underestimated denitrification in the Wadden Sea. On the one hand, this might artificially reduce the atmospheric nitrogen shares because they should rise in periods of nitrate depletion. On the other hand, $DIN_{atmos}$ should also be strongly depleted as the remaining DIN when denitrification was working properly. Thus, the relative $DIN_{atmos}$ shares should remain on a similar level as presented here. It was a different situation if DIN was not denitrified and

released as $N_2$ but rather consumed by phytoplankton.

Figure 5 shows the total concentrations of DIN, PON, and chlorophyll-a averaged over the upper five model layers. It is though as reference for the comparison of the relative contribution of total, shipping-related, and agricultural-related nitrogen deposition to these tracers in Fig. 6 (also top three model layers). Hence, Fig. 5 is not further discussed in this context.

### 3.2.2 The shipping sector

The relative $DIN_{ship}$ share is highest in regions of the Baltic Sea, which are distant to anthropogenic land-based activities – namely the Eastern Gotland Basin and the Bothnian Sea – but also in the western Baltic Sea (Fig. 6). Though, it is quite low with $3\%$ to $4\%$ at the considered stations in these regions (Table 3) but may reach values above $5\%$ further north. Particularly these regions distant to anthropogenic land-based activities are less impacted by other nitrogen deposition source such as agricultural activities. Hence, shipping-activities are a relevant atmospheric nitrogen source in less populated regions of the Baltic Sea.

Particularly in summer the $DIN_{atmos}$ shares take similar values as $DIN_{agri}$ shares of more than $10\%$ at BY15 – nearly $20\%$. These higher shipping contributions in summer are known facts and consistent with results of other studies (Aulinger et al., 2016; Raudsepp et al., 2013).

The $DIN_{ship}$ contributions are higher than $Chl\text{-}a_{ship}$ contributions per individual month. On annual average, however, the $PON_{ship}$ shares are slightly higher than $DIN_{ship}$ shares and the $Chl\text{-}a_{ship}$ shares are slightly higher than $PON_{ship}$ shares.

Thus, we see a slight enrichment of shipping related nitrogen in biota. In the southern North Sea the $DIN_{ship}$ share is quite constant with approximately $1\%$, whereas in the northern North Sea it takes values of up to $3\%$. These lower shares compared to the Baltic Sea do not mean that the absolute shipping-related nitrogen deposition is lower in the North Sea. Instead, the



absolute DIN concentrations are higher in the North Sea leading to lower $DIN_{ship}$ shares. The $PON_{ship}$ and Chl-$a_{ship}$ shares take similar values.

Shipping is a ubiquitous source of $NO_X$ and particle emissions in marine regions, which is higher along the densely frequented shipping routes. But $NO_X$ and fine particles do not deposit close to their sources. As can be expected from the map of the shipping-related nitrogen deposition (Fig. 3, left), it is not higher close to shipping routes than farther away. Therefore, homogeneous $DIN_{ship}$, $PON_{ship}$ and Chl-$a_{ship}$ concentrations are reasonable. The shares are not homogeneous in space because DIN concentrations are higher along the coast than at the open sea or higher in the North Sea than in the Baltic Sea.

The atmospheric DIN shares and particularly the $DIN_{ship}$ share rise in summer. DIN is depleted during this time of year and less DIN is provided from other sources, which indirectly increases the atmospheric DIN shares. Additionally, the atmospheric concentrations of $NO_X$ and their deposition are higher in summer (Aulinger et al., 2016; Raudsepp et al., 2013). On the one hand, this nitrogen is bad for the ecosystem because it further pushes the phytoplankton growth. On the other hand, it is good nitrogen in the Baltic Sea because it increases the N:P ratio and fosters diatom and flagellate growth. Thus, the excess dissolved inorganic phosphorus decreases leading to a reduction of cyanobacteria blooms (Wasmund, 1997; Niemi, 1979; Janssen et al., 2004). Raudsepp et al. (2013) showed for the Gulf of Finland that shipping-related nitrogen deposition led to a reduction of nitrogen fixation by $2\%$ to $6\%$. In this context, it was very important to also consider the spatio-temporal pattern of phosphorus deposition.

The North Sea and Baltic Sea will be declared as nitrogen oxide emission control areas (NECAs) from 2021 and onwards. Because only new build ships are affected by the NECA Tier III emission thresholds, it will take several years until positive effects will become visible. Hammingh et al. (2012) estimated for 2030 that approximately $6\%$ of the nitrogen deposition into the North Sea would originate from the shipping sector without NECA Tier III emission thresholds. Compliance with these thresholds would reduce the shipping contribution by 1/3 to $4\%$. Considering, that only $1\%$ to $1.5\%$ of the DIN in the North Sea originates from shipping, a $33\%$ reduction of the shipping input would have a minor impact on the nitrogen budget. Transferring this $33\%$ reduction to the Baltic Sea might have relevant positive effects in its western part, which is generally heavily impacted by anthropogenic nitrogen inputs. As indicated by the study of Raudsepp et al. (2013), a reduction of shipping emissions might have negative impacts in other regions of the Baltic Sea fostering cyanobacteria blooms by reducing the N:P ratio.

The North Sea and Baltic Sea are sulfur emission control areas (SECAs) and since January 2015 a threshold of $0.1\%$ (m/m) sulfur in fuel is in force. Hence, the shipping-related $SO_2$ emissions and the formation of $H_2SO_4$ have been considerably reduced in recent years. Lower $NO_X$ emissions in NECs will lead to less $HNO_3$. Thus, less acidic compounds are available in the marine atmosphere since 2015 and their availability will even further decrease after 2020 increasing the pH value of atmospheric particles. Less $NH_3$ will condense onto particles and capture a proton because this process is favored by low particle water pH values (e.g. Fig. 1 in Neumann et al. (2018b)). Probably this shift from $NH_4^+$ to $NH_3$ will change the deposition patterns of reduced nitrogen.





### 3.2.3 The agricultural sector

The agricultural ammonia emissions were found to contribute up to nearly $30\%$ to the DIN surface water pool in the western Baltic Sea in spring 2012 (Fig. 6). On annual average it is still approximately $11.5\%$. In the central Baltic Sea agricultural emissions contribute nearly $10\%$ and in the southern North Sea $2\%$ to $5\%$ to the DIN pool. With respect to the total atmospheric

nitrogen deposition they make up $30\%$ to $50\%$. Heavily agriculturally used areas are located upwind to the western Baltic Sea. Reasonably, the $DIN_{agri}$ shares are higher in this marine region compared to the Gotland Basin. This spatial pattern is also clearly visible when one compares the $DIN_{ship}$ and $DIN_{agri}$ shares in Fig. 6.

The $PON_{agri}$ and Chl-$a_{agri}$ shares are similar to the $DIN_{agri}$ shares at all stations and in the North Sea (Table 3). On Baltic Sea average, however, the $PON_{agri}$ and Chl-$a_{agri}$ shares are nearly $50\%$ higher. This has been also observed for the shipping

sector contribution.

The agricultural nitrogen emissions might have been underestimated as noted in the Materials and Methods section (Sect. 2.1). Briefly, the reasons are:

  a) Agriculturally emitted oxidized nitrogen compounds were not tagged in ERGOM because their share of the total oxidized nitrogen deposition could not be quantified.

b) Secondary agricultural emissions, such as wind-blown dust, were not considered in the emission data.

Hence, the actual agricultural contribution to marine DIN, PON, and chlorophyll-a pool is expected to be even a bit higher than $5\%$ to $10\%$ in the North Sea and Baltic Sea. Agricultural activities are an important contributor to nitrogen input into the ocean via the atmosphere Therefore, measures to reduce agricultural emissions could have higher impacts than reduction measures for shipping emission.

Kalli et al. (2010) estimated that abatement costs per t emitted N to reduce $NO_X$ emissions of ships are in a similar order of magnitude than costs to reduce agricultural riverine N release into the Baltic Sea. Only a proportion of shipping-emitted $NO_X$ deposits into the ocean. In contrast, agricultural riverine N emissions predominantly go directly into the ocean. Hence, 1 t of shipping emitted $NO_X$-N leads to less DIN than 1 t of agricultural nitrogen released via rivers (which is 1 t). In contrast, a large proportion of riverine nitrogen is denitrified in flat coastal regions not reaching the central basins where atmospheric

nitrogen deposition is relevant (Voss et al., 2010). Hence, it is difficult to weigh up the abatement costs for atmospheric shipping emissions and for riverine agricultural inputs.

### 3.3 Comparison to other Studies

Große et al. (2017) evaluated the impact of different DIN sources on the oxygen deficit in the North Sea from 2000 to 2014. He also considered atmospheric nitrogen deposition and presented its share to total nitrogen (TN) in the water column (Große

et al., 2017, Fig. 6). The spatial pattern of the $TN_{atmos}$ shares in Große et al. (2017) is similar to pattern of the $DIN_{atmos}$ shares in this study (Fig. 6, top left) but the values are slightly higher. However, there is a spatial hotspot in the $TN_{atmos}$ share westward of the Rhine estuary, which does not arise in this study's $DIN_{atmos}$, $PON_{atmos}$, or Chl-$a_{atmos}$ shares.



Los et al. (2014) considered the contribution of various sources to the winter DIN concentrations in the North Sea (Los et al., 2014, Fig. 5). They found highest $DIN_{atmos}$ shares westward of Denmark (up to approximately $20\%$), which decreased in the German Bight – approximately $10\%$ to $15\%$ in the southern German Wadden Sea and $5\%$ to $10\%$ elsewhere – and further decreased further northwestward and towards the English Channels. These numbers and pattern also roughly agree with our results.

Troost et al. (2013) evaluated the contribution of atmospheric deposition to total nitrogen ($TN_{atmos}$) for the year 2002. They compared three methods to evaluate the contribution of specific sources: the tracer based method, a budget based method, a removed based method. The first method was also used in this study. The latter method was used in the CTM simulations with CMAQ to calculate the shipping contribution. Troost et al. (2013) found highest $TN_{atmos}$ shares in the central North Sea of more than $20\%$ (Troost et al., 2013, Fig. 3.1), which is higher than in this study. But, they decreased considerably in southward direction towards the coast of Belgium, the Netherlands, and German. $TN_{atmos}$ shares fall below $10\%$, which is lower than the $DIN_{atmos}$ shares in this study. The inflow from the English Channel or from the Rhine was possibly lower in the year 2002 than in 2012 leading to the described difference along the coastline. Troost et al. (2013) further derived average $TN_{atmos}$ and $Phytoplankton_{atmos}$ shares of $6\%$ and $7\%$, respectively, for the southern North Sea. These values are lower than the $DIN_{atmos}$ and $Chl\text{-}a_{atmos}$ shares at the three North Sea stations in this study, which amount $7\%$ to $10\%$.

Ménesguen et al. (2018) considered the Bay of Biscay and ends northeastward of the English Channel and Dulière et al. (2017) considered only the English Channel. Therefore, their results are difficult to compare to this study. Dulière et al. (2017) found that $10\%$ to $30\%$ of the nitrogen contributed to English Channel originated from the atmosphere in the years 2000 to 2010.

The studies of Große et al. (2017), Los et al. (2014), and Troost et al. (2013) used annual or monthly averaged EMEP nitrogen deposition data. The generally slightly higher shares of $DIN_{atmos}$ observed in Große et al. (2017) might be caused by the fact that the EMEP nitrogen deposition is nearly $50\%$ higher than the CMAQ nitrogen deposition used in this study (see Sect. 3.1 Neumann et al., 2018b). A comparison of CMAQ and EMEP nitrogen deposition as forcing for HBM-ERGOM simulations in the western Baltic Sea also showed that EMEP deposition leads to higher DIN concentrations. Vivanco et al. (2017) compared the nitrogen deposition of EMEP, CMAQ, and four further models. EMEP deposition was closest to measurements whereas CMAQ deposition indicated considerable underestimations.

In contrast, Los et al. (2014) and Troost et al. (2013) found lower atmospheric nitrogen shares in the surface water. Different years are considered, in which the atmospheric contribution might have been lower, were considered in the latter two studies. This discrepancy might also be related to higher riverine nitrogen inputs. Because the denitrification in the southern North Sea is underestimated in this study we do not compare absolute DIN concentrations of these studies.

The only study in the Baltic Sea region, which goes beyond nitrogen deposition estimates, is Raudsepp et al. (2013). They performed two ecosystem model simulations with GETM-ERGOM in the Gulf of Finland: one with the total nitrogen deposition and one without shipping nitrogen deposition. They evaluated the impact of shipping-related nitrogen deposition on the N:P ration and on nitrogen fixation by cyanobacteria. However, they did not provide figures, to which we can compare our $DIN_{ship}$ or $Chl\text{-}a_{ship}$ estimates.





A comparison of our results for the whole Baltic Sea with other studies is currently not possible due to a lack of them. Similar model simulations as done in this study were performed within the EU BONUS Project SHEBA. A publication is in preparation but not submitted yet.

## 4 Conclusions

An HBM-ERGOM model simulation for the year 2012 with tagged total atmospheric, shipping-related, and agricultural nitrogen deposition was evaluated in this study. The surface water concentrations averaged over the top five model layers (approx. top 12 m) were considered for this purpose. The evaluated year was preceded by four consecutive iterations of the same year as spin-up phase for the tagged tracers. The nutrient concentrations at the end of one iteration were used as initial conditions for the next iteration except for silicate, which was restored to its initial conditions each year. The first year of the tagging-spin-up was used for the model validation presented in part A of this study (Neumann et al., 2018b).

Atmospheric nitrogen deposition contributed $12\%$ to $13\%$ to the DIN-N in the North Sea and Baltic Sea on annual average. The atmospheric contribution was lower at the three stations in the southern North Sea ($7\%$ to $10\%$) but considerably higher at the two stations considered in the Baltic Sea ($20\%$ to $30\%$). The discrepancy of high $DIN_{atmos}$-shares at the two Baltic Sea stations compared to the Baltic Sea average has two reasons: (a) riverine nutrient input dominates along the coast line but the two stations are in open waters; (b) the atmospheric nitrogen deposition decreases in the farther northward and eastward Baltic Sea regions, increasing the relevance of nitrogen fixation by cyanobacteria.

In the North Sea, the relative $PON_{atmos}$ and $Chl\text{-}a_{atmos}$ shares were slightly lower than the $DIN_{atmos}$ shares: $10\%$ to $11\%$. The fact that the $DIN_{atmos}$ shares are higher than the $PON_{atmos}$ shares results from the annual flushing of the southern North Sea. This flushing does not allow nutrients to accumulate over several years as described in Neumann et al. (2018b).

The inverse situation prevails on average in the Baltic Sea: the relative $PON_{atmos}$ and $Chl\text{-}a_{atmos}$ shares were higher than the $DIN_{atmos}$ shares taking values of $18\%$ and $19\%$, respectively. To explain this situation, the offshore areas – particularly the Baltic Proper, the Arkona Sea, and the Bothnian Sea – and coastal areas need to be considered individually. In the offshore areas the relative $DIN_{atmos}$, $PON_{atmos}$, and $Chl\text{-}a_{atmos}$ shares are high ($20\%$ to $35\%$) and nearly equalized compared to each other (Fig. 6; columns OMBMPK8 and BY15 in Table 3). In contrast, these relative shares are very low in many coastal areas giving the impression that these areas contain less atmospheric nitrogen than the offshore areas. But, this is not correct: the absolute atmospheric nitrogen input (Fig. 2) and the absolute DIN concentrations (Fig. 5) are very high. The relative atmospheric nitrogen shares are low because nitrogen input by rivers is even higher than the atmospheric nitrogen input. Simultaneously, the algal growth is not limited by bioavailable nitrogen but by bioavaiable phosphorus in the river plumes yielding high DIN concentrations. Resulting, the relative $DIN_{atmos}$ share is low in many coastal areas. But, the coastal DIN concentrations considerably influence the spatial average $DIN_{atmos}$ shares of the whole Baltic Sea, by having high absolute $DIN_{atmos}$ concentrations along the coast.

Other recent studies also dealt with the contribution of atmospheric nitrogen to the nitrogen concentrations in the North Sea (Große et al., 2017; Troost et al., 2013; Los et al., 2014). Those studies' atmospheric nitrogen shares were in the same order of





magnitude. The spatial patterns were similar. However, the maximum atmospheric nitrogen shares exceeded the highest shares in this study. Annual or monthly mean nitrogen deposition data of the EMEP model were used in most studies. The comparison between the CMAQ and EMEP nitrogen deposition data in the part A of this study (Neumann et al., 2018b) showed that the annual average EMEP nitrogen deposition of the North Sea is approximately $50\%$ higher. Vivanco et al. (2017) also compared EMEP and CMAQ nitrogen deposition and had similar findings. Based on these comparisons it is reasonable that model experiments, which use EMEP data, yield higher atmospheric contributions than the model experiments in this study. This was also shown by another modeling study covering the western Baltic Sea, which was recently submitted (Neumann et al., 2018a). Different atmospheric deposition data sets should be used in future studies to evaluate the sensitivity of the ecosystem on nitrogen deposition. The most natural choice would be to use EMEP nitrogen deposition data as a first data set for such a comparison because it is available for a time span of more than 15 years.

On annual average, the agricultural emission sector made up $30\%$ to $50\%$ of the total atmospheric nitrogen deposition and contributed $5\%$ to $10\%$ to the DIN pool depending on the region. The agricultural emissions are afflicted with considerable uncertainty due to rough spatio-temporal ammonia emission profiles and missing information on soil and crop types. This uncertainty in the emissions affected the agricultural nitrogen deposition. The contribution through agricultural nitrogen emissions probably has been underestimated because agricultural oxidized nitrogen and wind-blown dust emissions were not marked in the nitrogen deposition data.

The relative $DIN_{agri}$ share of $5\%$ to $10\%$ shows that a substantial nutrient input by agricultural activities is transported into the oceans not only via rivers but also via the atmosphere. Thus, measures to reduce livestock density would not only reduce riverine nitrogen inputs but would also reduce atmospheric reduced nitrogen deposition by up to $50\%$ in some regions. This could decrease the marine DIN pool by up to $10\%$ in the long run. Lower ammonia emissions would also impact atmospheric particle formation and, hence, feed back on sulfate and nitrate deposition to the ocean. Detailed estimations on ammonia emissions and on emissions reductions for different manure application and meat consumption scenarios have been published in recent years (Skjøth et al., 2004; Skjøth et al., 2011; Backes et al., 2016b; Hendriks et al., 2016). Based on these studies, more detailed assessments of the impact of agricultural activities and different technologies on the atmospheric nitrogen input into marine water bodies could be performed in future research.

The shipping contribution was very low but spatially constant in the North Sea and Baltic Sea region taking values of approximately $1\%$ and $3\%$, respectively. Thus, the contribution of the shipping sector is less relevant than the agricultural sector. In cases of strong reduction of agricultural nutrient input via rivers and the atmosphere, the relevance of the shipping sector will rise. In addition in future studies, the shipping-related nitrogen deposition should be compared to the deposition of nitrogen from land-based anthropogenic $NO_X$ emission sectors, such as energy generation and road transport. Their contribution to the marine DIN pool and marine biomass should be evaluated. It has to be recognized that emission reduction measures in the shipping sector like Nitrogen Emission Control Areas (NECAs) will have a low impact on the marine nitrogen budget of the North Sea and Baltic Sea even if they reduce the shipping related nitrogen deposition by approximately 1/3 (Hammingh et al., 2012). However, this might be different in individual regions.



The nitrogen-to-phosphorus ratio is very high in shipping emissions compared to riverine inflow from agricultural sources. In this model study, shipping emissions do not even contain phosphorus. In the Baltic Sea, high N:P ratios favor the growth of diatoms and flagellates, whereas a low N:P ratio favors the growth of cyanobacteria because high excess DIP concentrations are left unmetabolized by diatoms and flagellates (Wasmund, 1997; Janssen et al., 2004). Hence, shipping-related nitrogen deposition has a positive impact by indirectly inhibiting the growth of cyanobacteria (Raudsepp et al., 2013).

Implicitly, this points to another issue: the knowledge on spatio-temporal phosphate deposition is low and needs to be improved (e.g., Mahowald et al., 2008; Rolff et al., 2008). Until more continuous measurement time series are available (e.g., HELCOM, 2017), sensitivity model simulations on phosphorus deposition could be performed. However, these sensitivity simulations should cover a few decades as part A of this study indicated.

Generally, long term simulations are necessary to properly evaluate the effects of changes of atmospheric deposition – such as effects of NECAs – because it takes several years until changes in deposition take their full effect in the Baltic Sea region as this study indicates. Moreover, the contribution of other anthropogenic emission sources to atmospheric nitrogen deposition should be considered in future studies.

*Code and data availability.* .

**Model Code:** The original HBM-ERGOM code was provided by the Federal Maritime and Hydrographic Agency of Germany (BSH). The license agreement does not allow the authors to pass the code to third parties. The code can be requested from the BSH or the Danish Meteorological Institute (DMI). The modified ERGOM code and brief description of the model processes and constants are attached in the supplement.

**Model output data:** The data are available via the THREDDS server of the IOW: https://thredds-iow.io-warnemuende.de/thredds/projects/meramo/catalog_meramo_cmaq16_silrestart.html

**Measurement data:**

– HELCOM data are available via the ICES homepage: http://ocean.ices.dk/helcom/Helcom.aspx

– IOWDB data are available on request (https://www.io-warnemuende.de/iowdb.html). Please contact to authors to get access to the database.

– ICES data are available via the ICES Oceanography data database http://ocean.ices.dk/HydChem/HydChem.aspx

– DOD data are available on request via the DOD Cruise Data Mining portal http://seadata.bsh.de/csr/retrieve/dod_index.html

*Author contributions.* .

**Daniel Neumann:** overall structure; HBM-ERGOM model simulations; programming work; plotting; major writing tasks

**Hagen Radtke:** implementation of the tagging method; framework for data processing; contribution to Results & Discussion and Materials & Methods sections; discussions during data evaluation; development of research question

**Matthias Karl:** CMAQ air quality model simulations; evaluation of meteorological forcing data and of nitrogen deposition data; contribution to Materials & Methods and Results & Discussion sections; development of research question



**Thomas Neumann:** development of the research question; contribution to Introduction, Materials & Methods and Conclusions

*Competing interests.* The authors declare that they have no conflict of interest.

*Acknowledgements.* Parts of the research published in this publication were carried out in the research projects MeRamo (funded by BMVI, FKZ 50EW1601) and BONUS SHEBA (Sustainable Shipping and Environment of the Baltic Sea region). The BONUS SHEBA project was supported by BONUS (Art 185), funded jointly by the EU and national funding institutions. The HBM-ERGOM model simulations were performed at the cluster Gottfried of the North-German Supercomputing Alliance (HLRN, project ID mvk00054). The meteorological and atmospheric chemistry transport model (CTM) simulations were performed at the German Climate Computing Center (DKRZ) within the Project "Regionale Atmosphärenmodellierung" (Project ID 0302), which is funded by the Helmholtz Association. The emissions for the air quality model simulations were kindly provided by Johannes Bieser, Armin Aulinger, and Jukka-Pekka Jalkanen. The HBM is currently maintained by the Danish Meteorological Institute (DMI) and the Federal Maritime and Hydrographic Agency of Germany (BSH), namely Thorger Brünning. The air quality model CMAQ is developed and maintained by the U.S. Environmental Protection Agency (US EPA). We thank our colleagues conducting IOW's Baltic Monitoring and long-term data program, which intense quality checked measurements we used for the model validation. Some of the measurement data were kindly provided by the HELCOM oceanographic measurements database hosted by ICES. Martin Schmidt of the IOW supported us with respect to preparation and upload of model data to the IOW THREDDS server. Anja Eggert commented the plotting and provided information on phytoplankton growth. Johannes Pätsch provided value input on the residence time of nutrients in the German Bight. We thank Uwe Schulzweida, Charlie Zender, Paul Wessel, the R Core Team, and the Unidata development team (and all involved developers/contributors) for maintaining the open source software packages Climate Data Operators (cdo), the NetCDF Operators (NCO), Generic Mapping Tools (GMT), the statistical computing language R, and netCDF, respectively.





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
