# Peer review of "Evaluation of atmospheric nitrogen inputs into marine ecosystems of the North Sea and Baltic Sea – part B: contribution by shipping and agricultural emissions"

_Biogeosciences, 2018_

## Referee Comment (RC1) · Anonymous Referee #5 · 23 Nov 2018

Review of the manuscript "Evaluation of atmospheric nitrogen inputs into marine ecosystems of the North Sea and Baltic Sea - Part B: contribution by shipping and agricultural emissions" by D. Neumann, H. Radtke, M. Karl and T. Neumann

SUMMARY:

The authors investigate the fate and behaviour of atmospheric nitrogen deposition form shipping and agricultural activities in the North and Baltic Sea. The study is based on a tagging method in the coupled physical-biogeochemical model HBM-ERGOM. Re-

gional fractions of atmospheric nitrogen are provided for inorganic nitrogen, particulate organic nitrogen and chlorophyll-a.

MAJOR COMMENTS:

While I think it is important to investigate the impact nutrient inputs related to shipping and agriculture on the Baltic and North Sea ecosystems, I must admit that I got lost in the description of many details and had problems to identify a clear aim. In the given context, I would mostly be interested in ecosystem changes due to atmospheric nutrient deposition and thus rather expect something like sensitivity experiments with and without this extra nutrient supply. I am not sure what to gain from tagging the fraction of atmospheric nitrogen shares in % to DIN, PON, and chlorophyll-a after five years. Another major point of criticism is the negligence of the strong impact of phosphorus. In the presence of nitrogen fixers, I regard the availability of phosphate as key. As I understood it, the phosphate input was set to a fixed value of unknown origin.

I am afraid that, in the present form, I have to reject the manuscript. I must, however, admit that I struggled to keep overview and it might well be that I missed an important point. I might thus change my mind, in case the authors could clarify their aim and the argumentation was more stringent.

SPECIFIC COMMENTS:

2 Materials and Methods

2.1 Atmosphere: I repeatedly lost overview. I would find it helpful if there was a more clear separation of the model assumptions, the input and the outcome. Also I would expect at least some evaluation of the results (apart from a non-published reference). While the authors state that everything is rather uncertain, they do not put this uncertainty into perspective. How do the modelled numbers compare the official estimates by HELCOM and OSPAR?

2.2 Ocean: Again, I find the model description confusing. Specifically, it did not get

clear to me why the simulation time was five years only (while the model is drifting?) and, also, it should, at least briefly, be mentioned how the key processes which impact the distribution of nutrients are implemented. Also the initial conditions of the model need to be clarified and I had problems to see why the physical model was restarted from its initial conditions (which?) each year. In addition, the model description would strongly gain from a comprehensive, clearly arranged list of nutrient sources and sinks in the model (e.g., is there a sediment model and burial? how large is the riverine input?). How did the authors determine the nitrogen fraction of chlorophyll a? Why did the authors chose to show atmospheric nitrogen shares in % to DIN, PON, and chlorophyll-a and which depth level do they consider, why?

Most important, however, I am not even sure what exactly was tagged – was the atmospheric deposition marked continuously or did the authors follow a pulse? In both cases there ratio behind the approach needs to be clarified.

Results - subsection 3.2: This section consists mainly of a list of numbers in % showing atmospheric nitrogen shares in % to DIN, PON, and chlorophyll-a (without providing any absolute values). Often I was not sure which region/depth levels the authors exactly refer to. Also, I lack explanations about reasons and ecological consequences (e.g., which paths did the nutrients take?). The few explanation provided did not become clear to me (e.g., why should offshore and coastal differences in the Baltic be explained by high DIN loads at the coast and P limitation?).

Results - subsections 3.2.2 and 3.2.3: Again, I found it very hard to keep overview. I would suggest to condense these parts considerably. Also I propose to focus more on the results and not to elaborate on the pros and cons of extra nitrogen input in general. Comparisons to the results of other studies could be summarized in a Discussion.

Conclusions: Also this Section would benefit from some guidance by the authors what the results mean for the ecosystem. As I see it now, it's mainly a repetition of the foregoing.

---

## Referee Comment (RC2) · Anonymous Referee #3 · 28 Nov 2018

This manuscript describes a model-based assessment of the impact of atmospheric nitrogen emissions from the shipping and agricultural sectors on the marine ecosystems of the Baltic and North Seas. The subject matter is interesting and appropriate for the journal. I have very few scientific comments related to the work, but I found the manuscript to be unnecessarily long and very repetitive. Reducing the length of the text and correcting the many typographical errors will help to make the manuscript more accessible and increase its impact.

Specific Comments: Reactivity of ammonia with sea salt: It is stated several times

that condensation of ammonia on seasalt particles enhances the removal of ammonium from the atmosphere in the coastal zone. Please could some specific citations of observational reports be added that support this statement? (Some literature is cited at the first occurrence (line 24, page 2), while subsequent statements do not include citations. Of the sources cited on page 2, the observational data presented in Kelly et al. 2010 (Figure 5 of the paper) directly contradict the statement). What mechanism drives this process? Length of manuscript and repetition: There are many examples where sections of text are repeated in this manuscript. I can see no advantage to this. I list some examples, but there are more. I would encourage the authors to remove as much repetition as possible from the manuscript. Examples: Page 10, line 1; P 11, L 16-17; P 17 L 17-20; P 17 L 24-26; P 18 L 11-15. In addition, I suggest that the following paragraph (P 3, L 27-34) is removed, since a) phosphorus is not the focus of the manuscript, b) the manuscript is already very long and c) there are other statements that address the limitations associated with phosphorus in the context of the manuscript.

P2, L22: I think the end of this line should be "at sea and partly deposit" P2, L30: Change "contribution to the oxidized nitrogen" to "contribution of the oxidized nitrogen". P3, L25-26: What is the "19%" a percentage of? P4, L6: delete the comma after "both". P4, L26 & L34: add an apostrophe after "sectors". P5, L12: Sentence should start "The dry deposition…" P5, L22: Insert "for" after "used". P5, L26: change to "considered to be the". P5, L29: Insert "the" before "difference". P6, L1: add a comma after "emission". P6, L3: change "word" to "work". P6, L5: Grid is 16 x 16 km, not km2. P7, L11: insert "by" after "applied". P7, L21: "spun" not "spin". P8, L19: I'm not sure whether the description of Section 3.2 is intended to be humorous, but it is subjective and inappropriate. P9, L13: "coastal" not "coastline". Is it really necessary to include this long quotation from the companion paper? P10, L8-9: Change end of sentence to "emissions from other sectors." P10, L 12: Change start of sentence to "The shipping sector…" P10, L14: Change to "found a relative contribution of". P11, L2: "agricultural" P11, L7: delete "by". P11, L22: insert space after "1/3". P12, L5: Change to "remains

high during summer but decreases" P12, L 9: insert "proportionally" after "contributes". P14, L6: insert "such" after "rivers". P14, L7: Change to "share than is the case" P16, L11: please explain the remark about denitrification in the Wadden Sea and add a reference. P16, L16-17: "It is though as reference for the comparison". I'm afraid I do not understand this phrase. P16, L20: delete comma after "Baltic Sea". P16, L23: Change to "deposition sources such as". P17, L29: Change to "NECAs"? P18, L20-26: I do not think that the authors have established why this paragraph contains a comparison to riverine inputs. What about measures to reduce agricultural emissions to the atmosphere? P19, L4: Change to "English Channel". P19, L11: Change to "Germany". P19, L28: delete "are considered". P19, L31: delete comma after "region". P19, L34: "ratio", not "ration". P21, L3: Change to "in part A". P22, L1-2: It is not clear to me why the authors specifically discuss the N:P ratio of shipping emissions here, when other emission sources are not discussed. Why is this introduced for the first time in the Conclusions section? P22, L7: Change "Until" to "Once".

---

## Referee Comment (RC3) · Savchuk (Referee) · 6 Dec 2018

Reviewer's comments to manuscripts by Neumann, D., Karl, M., Radtke, H., and Neumann, T. "Evaluation of atmospheric nitrogen inputs into marine ecosystems of the North Sea and Baltic Sea – part A: validation and time scales of nutrient accumulation; part B: contribution by shipping and agricultural emissions" submitted to "Biogeosciences"

The study aims at a detailed quantitative description of the pathways and effects of

atmospheric nitrogen inputs in the marine ecosystems of the North and Baltic seas as simulated with the coupled physical-biogeochemical model HBM-ERGOM. The sine qua non precondition for achieving such ambitious, if somewhat artificial, goal is the realistic simulation of biogeochemical nitrogen cycling in both marine systems. That's why both manuscripts must be considered together, starting from the model itself. Unfortunately, the implemented model version is not suitable for such studies in many aspects: A) by deficient formulations; B) by failing in reproducing some phenomena crucially important in nitrogen cycling; C) by flawed set-up of numerical experiments and validation; and, finally, D) by poor model-data comparability. All these, taken together convert presented results in merely casual exercises that have little to do with the realistic cycling of atmospheric nitrogen in marine ecosystems. That's why I would not even go further into detailed reviewing of "tagged" results. Instead, I recommend to reject both manuscripts and advice against using this version of HBM-ERGOM model, made for operational purposes (perhaps, with the data assimilation), for the long-term studies.

A few examples of crucial flaws and drawbacks are given below.

A) "Iron reduction and release of phosphate under anoxic conditions in the sediment are not represented in this ERGOM version" (Part A, L 15/8). Fixing sediment N:P ratio and ignoring redox alterations of the P cycle implausibly affects phosphate dynamics, hence, distorts such important flux as nitrogen fixation and the following cycling of fixed nitrogen. The necessity of Si restarting for every year indicates that its dynamics even during the first iteration is erroneous with corresponding consequences for phytoplankton seasonal succession and nutrient uptake. Finally, many important features and phenomena, for instance, nutrient limitation, nutrient residence times, species composition, tides and oceanic impacts, etc., are rather different between the North and Baltic seas. That makes combining them into a single domain questionable, if not harmful for the objectives of this study.

B) Overestimated deep layers oxygen concentration and underestimated denitrification

distort DIN distribution and dynamics (see comparisons in Figs. 7-10). Together with questionably reproduced nitrogen fixation, such underestimation indicates a wrong balance between nitrogen sources and sinks, hence, biases evaluation of atmospheric N contribution to unknown degree.

C) "Therefore, a detailed validation of the nitrogen deposition data sets is not possible and it is not clear whether the CMAQ nitrogen deposition is actually too low over sea." (Part A, L16-18/13). Already this statement makes studies of the RELATIVE contributions rather uncertain. Further uncertainty (due to possible non-linear effects in the biogeochemical cycling) is introduced by the repetitive implementation of deposition computed only for one year (i.e. 2012) over all five years, forcing a possible deficit accumulation.

D) The model set-up and simulated dynamics contain many features that are "typical within order of magnitude" rather than year-specific. Therefore a comparison of the "first" iteration with observations during concrete 2012 year looks very optimistic, even naïve. Perhaps, such choice partly explains why most patterns of seasonal dynamics are very poorly reproduced either in timing or by the levels, or both (Figs. 7-10). Never mind the plausible oxygen dynamics in the surface layer, where it is mainly driven by air-sea gas exchange. Moreover, the focusing of analysis at the surface layer is unwarranted because the nitrogen biogeochemical cycle must be evaluated for the entire ecosystem, including sediments.

---

## Referee Comment (RC4) · Anonymous Referee #1 · 8 Dec 2018

Neumann et al have attempted to estimate contribution of shipping and agricultural emissions to atmospheric nitrogen inputs into the North Sea and Baltic Sea. Their analysis is based on several models and most of these codes cannot be provided to the third parties, as mentioned by the authors. In such a situation, it is difficult to properly evaluate the findings in this study. In addition, there seems to some fundamental problems in the manuscript. I list my concerns one by one below:

1. Bioavailable PON has been mentioned at several places (such as line 21, page 1). This is a misleading term unless further qualified. Only inorganic nitrogen (dissolved

nutrients form) and up to certain extent, DON is considered bioavailable. I have not seen any (oceanographic) study, where PON is proposed to be bioavailable. I am unsure if the authors wanted to convey the availability of nitrogen to heterotrophs (such as fish), then PON can be bioavailable. But in traditional view, we do not present the definition of bioavailable in this way.

2. Authors must specify how does their model is able to differentiate between different components of anthropogenic (for that matter natural as well) inputs?

3. No uncertainties are provided in the estimates. It is important to provide uncertainties (in all tables and texts, wherever an estimate is quoted) anyways but here it more important as % contribution difference of different processes in the two basins is not much.

4. What deposition velocities are used in the model to estimate deposition rates? These must also have large uncertainties.

5. Chlorophyll is a pigment so how does one estimate relative contribution of shipping etc to chlorophyll and what does it signify (Fig. 6)? Perhaps an estimate to primary production instead of chlorophyll would have been meaningful.

6. Why a particular year (2012) is chosen (line 5, page 20)? Will the conclusions change for another year? How does one specify a particular year in model (unless there is some time-series analysis involved, which is not the case here)?

7. Baltic Sea is also zone of nitrogen inputs through N2 fixation. Is this component taken into account in the model?

Such studies are important to advance our understanding of anthropogenic nitrogen inputs to marine ecosystems but this study, unfortunately, does not tell any important story in that direction.

---

## Author Comment (AC1) · 12 Dec 2018

**Response to review comment #1 by referee #5**

We thank the reviewer for the constructive comments on the manuscript.

Below, the reviewers comments are written in bold letters and our answers in non-bold letters.

[Figure]

**Major comments**

**While I think it is important to investigate the impact nutrient inputs related to shipping and agriculture on the Baltic and North Sea ecosystems, I must admit that I got lost in the description of many details and had problems to identify a clear aim.**

**In the given context, I would mostly be interested in ecosystem changes due to atmospheric nutrient deposition and thus rather expect something like sensitivity experiments with and without this extra nutrient supply. I am not sure what to gain from tagging the fraction of atmospheric nitrogen shares in % to DIN, PON, and chlorophyll-a after five years.**

> We agree that it is reasonable to turn individual source sectors like shipping or agricultural activities on and off if one wanted to see how the system reacts on the reduction of anthropogenic emissions. This approach is reasonable for assessing the impact of emission reduction legislation. However, this studies deals with one step earlier: does (total/shipping-related/agricultural-related) nitrogen deposition contribute significant amounts of nitrogen to the marine biogeochemical processes and, if yes, where is it the case? Correspondingly, we formulated our research questions, i.e. "*What contribution do shipping- and agricultural-related nitrogen emissions [into the atmosphere] have [...] to the marine nitrogen concentrations?*". Hence, we considered % of total/shipping-related/agricultural-related atmospheric nitrogen to DIN, PON and chlorophyll. After identifying source sectors with relevant impacts, the next step would be to turn off these sectors in model studies.

> We will try to formulate the research questions and the introduction more concise in a revised version of the manuscript.

**Another major point of criticism is the negligence of the strong impact of phos-**

**phorus. In the presence of nitrogen fixers, I regard the availability of phosphate as key. As I understood it, the phosphate input was set to a fixed value of unknown origin.**

> We are pleased that the reviewer points to this aspect and we strongly agree with the reviewer that correct atmospheric phosphorus deposition is important.

> The fixed value for the phosphorus deposition was set to a value close to the currently by HELCOM suggested value. Someone, whom we could not identify, tuned the value of the phosphorus deposition. We know that this does not comply with the idea of good scientific practice and we are not happy with this situation.

> We are not aware of validated deposition fields of bioavailable phosphorus for the North Sea and Baltic Sea region. OSPAR does not consider this topic and HELCOM suggested one spatio-temporally constant value. We would have liked to stick to the approaches chosen in other model studies but phosphorus deposition commonly is not documented in the Material and Methods sections of respective publication.

> Most main-stream atmospheric chemistry transport models do not consider phosphorus compounds. First, phosphorus is not of high interest for atmospheric chemistry processes. Second, emissions of some source sectors are difficult to estimate: i.e. detailled soil information is necessary to properly calculate wind-blown dust emissions of phosphorus (soil type, phosphorus content, humidity, plant cover). Third, emitted phosphorus compounds are processed during their atmospheric transport and the fraction of bioavailable phosphorus to the time of deposition is difficult to assess.

[Figure]

**Specific comments**

**2.1 Atmosphere: I repeatedly lost overview. I would find it helpful if there was a more clear separation of the model assumptions, the input and the outcome.**

> We will restructure the section.

**Also I would expect at least some evaluation of the results (apart from a non-published reference).**

> We included a brief comparison with EMEP nitrogen deposition data in companion paper part A. The validation paper has recently been published as discussion paper in Atmospheric Chemistry and Physics as Karl et al. (2018, "*Impact of a nitrogen emission control area (NECA) on the future air quality and nitrogen deposition to seawater in the Baltic Sea region*", doi: 10.5194/acp-2018-1107).

**While the authors state that everything is rather uncertain, they do not put this uncertainty into perspective. How do the modelled numbers compare the official estimates by HELCOM and OSPAR?**

> We will restructure the section.

> To the best of out knowledge, the estimates of HELCOM and OSPAR are based on EMEP data. We compare EMEP and CMAQ nitrogen deposition in the companion paper part A. CMAQ nitrogen deposition is lower than EMEP nitrogen deposition according to recent comparison study of atmospheric chemistry transport models (Vivanco et al., 2017, doi: 10.1016/j.atmosenv.2016.11.042). Moreover, EMEP nitrogen deposition was closer to land-based nitrogen deposition measurements. Nitrogen deposition measurements above the North Sea and Baltic Sea are not available in reasonable spatial

or temporal resolution. Hence, one might assume that CMAQ underestimates the nitrogen deposition. We mentioned this in the manuscript (p.19, l.23–26) but should have put this information to another place in the manuscript. We will do it if we are allowed to submit a revised version.

**2.2 Ocean: Again, I find the model description confusing. Specifically, it did not get clear to me why the simulation time was five years only (while the model is drifting?) [. . . ]**

> We had two major reasons for considering five years. It is explained in companion paper part A. First, we were limited in the computing time when we performed the model simulations for the study. The convergence of the tagged atmospheric nitrogen towards a steady-state seemed to be sufficient after five years in most regions of the model domain except for the Gothland Basin. Second, we are aware of shortingcomings in our model setup with respect to the oxygen cycle and the denitrification in deep layers of the Gothland Basin (companion paper, Fig. 8). We hoped to avoid a feedback of these deep layers on the surface layer concentrations by keeping the simulation time as short as necessary.

**[. . . ] and, also, it should, at least briefly, be mentioned how the key processes which impact the distribution of nutrients are implemented.**

> We will add information on the transport processes if we are allowed to submit a revised version.

**Also the initial conditions of the model need to be clarified and I had problems to see why the physical model was restarted from its initial conditions (which?) each year.**

[Figure]

> The physical model is restarted each annual iteration from spun-up initial contions for the year 2012. We did this to be able to assess the biogeochemical model results without interferences by the physics: the physics receive no feedback from the biogeochemistry. Thus, the processes are the same in each iteration. Variations in the biogeochemistry from one iteration to the next are only due to biogeochemical processes and not due to variations in the ocean physics. We described it only in companion paper part A (p.9, l.17–20) and forgot to mention it in this part B.

**In addition, the model description would strongly gain from a comprehensive, clearly arranged list of nutrient sources and sinks in the model (e.g., is there a sediment model and burial? how large is the riverine input?).**

> The sediment is represented by one model layer at the bottom of the sea. It is mentioned in companion paper part A (p.8, l.16–17) but we forgot to describe it in this part B. We will add this information if we are allowed to submit a revised version.

> A detailed list of all processes is included in the Supplementary Material (pdf file starting with bm_ergom_2017_model_description*.

**How did the authors determine the nitrogen fraction of chlorophyll a?**

> We did not describe the usage of chloropyll properly. Chlorophyll is calculated as diagnostic variable from the phyoplankton concentration. Hence, the relative contribution of nitrogen source XY to chlorophyll actually means the quotient of *nitrogen of source XY in phytoplankton* divided by *total nitrogen in phytoplankton*. We decided to show and discuss chlorophyll because chlorophyll concentrations were validated.

**Why did the authors chose to show atmospheric nitrogen shares in % to DIN, PON, and chlorophyll-a . . . ?**

> Our major aim was to assess how relevant the contribution of total/shipping-related/agricultural-related atmospheric nitrogen deposition is for nitrogen compounds in the marine water compared to other sources. We found it reasonable to compare %-ages instead of absolute amounts.

**. . . and which depth level do they consider, why?**

> We presented the averaged over the top five model layers (approximately $12$ m; p.9, l.30–31). We know that the model has issues in the Gothland Basin (see our reply further above). We considered the upper $12$ m because we expect that this depth range is not impacted by deep water processes in the considered time frame.

**Most important, however, I am not even sure what exactly was tagged - was the atmospheric deposition marked continuously or did the authors follow a pulse? In both cases there ratio behind the approach needs to be clarified.**

> Atmospheric nitrogen was continuously tagged. It takes several years until atmospheric nitrogen reaches an equilibrium in the Baltic Sea. We wanted to assess how much atmospheric nitrogen deposition contributes to the North Sea and Baltic Sea in the long run. If we wanted to calculate the residence time of atmospheric nitrogen in the North Sea and Baltic Sea, an individual pulse would have been reasonable. We will add this information if we are allowed to submit a revised version.

**This section consists mainly of a list of numbers in % showing atmospheric nitrogen shares in % to DIN, PON, and chlorophyll-a (without providing any absolute values). Often I was not sure which region/depth levels the authors exactly refer to. Also, I lack explanations about reasons and ecological consequences (e.g., which paths did the nutrients take?). The few explanation provided did not be**

**come clear to me (e.g., why should offshore and coastal differences in the Baltic
be explained by high DIN loads at the coast and P limitation?).**

**Conclusions: Also this Section would benefit from some guidance by the au-
thors what the results mean for the ecosystem. As I see it now, it's mainly a
repetition of the foregoing.**

> We will clarify this and shorten the results section in a revised version of the
manuscript if we are allowed to submit a revised version.

––––––––––––––––––––––––––––

---

## Author Comment (AC2) · 12 Dec 2018

**Response to review comment #2 by referee #3**

We thank the reviewer for the positive feedback to our manuscript and for suggested improvements.

Below, the reviewers comments are written in bold letters and our answers in non-bold letters.

**Reactivity of ammonia with sea salt: It is stated several times that condensation of ammonia on seasalt particles enhances the removal of ammonium from the atmosphere in the coastal zone. Please could some specific citations of observational reports be added that support this statement? (Some literature is cited at the first occurrence (line 24, page 2), while subsequent statements do not include citations. Of the sources cited on page 2, the observational data presented in Kelly et al. 2010 (Figure 5 of the paper) directly contradict the statement). What mechanism drives this process?**

> Ammonia has a low residence time in the gas phase but tends to condense on surfaces (= ground or existing particles) or tends to form new particles. If there are relatively many large ("*coarse*") particles in the ambient atmosphere, relatively more ammonia condenses on these large particles. If there relatively many small ("*fine*") particles in the ambient atmosphere, relatively more ammonia condenses on these small particles. Coarse particles have a lower atmospheric residence time than fine particles because they have a higher dry deposition velocity (higher gravitational settling). Hence, if the major share of atmospheric particles is of coarse size, the dry deposition of ammonium is enhanced compared to a situation with predominantely fine particles. In coastal regions, sea salt is a dominant source for atmospheric coarse particles. Hence, a large fraction of ammonia condenses on the surface of sea salt particles, which desit fast to the sea.

> The reviewer is right, Kelly et al. (2010) does not state it directly. We mixed it with another publication. Please excuse us. Kelly just states: "*Due to the different deposition velocities of gases and particles, condensation of HNO 3 and NH 3 on coarse sea salt can alter nitrogen de position to sensitive ecosystems (Pryor and Sorensen, 2000; Evans et al., 2004).*". We will add further correct references if the paper is not rejected.

**Length of manuscript and repetition: There are many examples where sections**

**of text are repeated in this manuscript. I can see no advantage to this I list some examples, but there are more. [. . . ]**

> We will shorten the manuscript and consider the suggested improvements in a revised version of the manuscript.

———————————————

---

## Author Comment (AC3) · 12 Dec 2018

**Response to review comment #3 by Oleg Savchuk**

We thank the reviewer, namely Oleg Savchuk, very much for reading this and the companion manuscript. We agree with most of the four major critical aspects mentioned and cannot satisfy/disprove them completely.

Below, the reviewer's comments are printed in bold letters and our answers in non-bold

letters.

**[. . . ] Instead, I recommend to [. . . ] advice against using this version of HBM-ERGOM model, made for operational purposes (perhaps, with the data assimilation), for the long-term studies.**

> We will not use it again.

**A) "Iron reduction and release of phosphate under anoxic conditions in the sediment are not represented in this ERGOM version" (Part A, L 15/8). Fixing sediment N:P ratio and ignoring redox alterations of the P cycle implausibly affects phosphate dynamics, hence, distorts such important flux as nitrogen fixation and the following cycling of fixed nitrogen. The necessity of Si restarting for every year indicates that its dynamics even during the first iteration is erroneous with corresponding consequences for phytoplankton seasonal succession and nutrient uptake. [. . . ]**

> *no objection*

**[. . . ] Finally, many important features and phenomena, for instance, nutrient limitation, nutrient residence times, species composition, tides and oceanic impacts, etc., are rather different between the North and Baltic seas. That makes combining them into a single domain questionable, if not harmful for the objectives of this study.**

> Yes.

> The ERGOM was developed for the Baltic Sea and this version was adapted to also work in the North Sea. Particularly the dynamics of the biogeochemical system of the

North Sea are not properly reproduced by this ERGOM as one sees in the validation. As the Baltic Sea dynamics are reproduced considerably better then the North Sea dynamics – as one might expect because ERGOM was develop for the Baltic Sea –, we consider to remove the North Sea from this manuscript. There are still issues in the Baltic Sea model dynamics – silicate decline, simple sediment, and deep water oxygen – but we hope that the model quality in the Baltic Sea is sufficient for a publication of this study.

**B) Overestimated deep layers oxygen concentration and underestimated denitrification DIN distribution and dynamics (see comparisons in Figs. 7-10). Together with questionably reproduced nitrogen fixation, such underestimation indicates a wrong balance between nitrogen sources and sinks, hence, biases evaluation of atmospheric N contribution to unknown degree.**

> We think that the time scale of five years prevents badly predicted deep-layer oxygen concentrations to feed back to the surface layer nitrogen concentrations.

> We consider to remove the North Sea from the manuscript and only evaluated the Baltic Sea.

**C) "Therefore, a detailed validation of the nitrogen deposition data sets is not possible and it is not clear whether the CMAQ nitrogen deposition is actually too low over sea."(Part A, L16-18/13). Already this statement makes studies of the RELATIVE contributions rather uncertain. Further uncertainty (due to possible non-linear effects in the biogeochemical cycling) is introduced by the repetitive implementation of deposition computed only for one year (i.e. 2012) over all five years, forcing a possible deficit accumulation.**

> We agree that a bias in the nitrogen deposition will amplify further and further in the marine biogeochemical model in each annual iteration. However, the bias will also

amplify in a model study over five or ten consecutive years if the bias in the nitrogen deposition is systematic and not only present in the year 2012.

> Based on Vivanco et al. (2017), Karl et al. (2018), and companion paper part A, one can assume that EMEP nitrogen deposition is closer to reality than the CMAQ data of this study and that the used CMAQ setup for Europe (nearly the same in Vivanco et al. (2017) and here) tends to systematically underestimate nitrogen deposition. However, we are not aware of any nitrogen deposition data set, which is validated above the ocean in detail because (operational) continuous nitrogen deposition measurements are missing (or other nitrogen deposition measurements with sufficient temporal or spatial coverage).

> Most marine biogeochemical model studies do not mention the uncertainty of the driving nitrogen deposition data at all. Moreover, some studies use monthly or annual mean nitrogen deposition as input and re-distribute it onto individual days.

**D) The model set-up and simulated dynamics contain many features that are "typical within order of magnitude" rather than year-specific. Therefore a comparison of the "first" iteration with observations during concrete 2012 year looks very optimistic, even naïve. Perhaps, such choice partly explains why most patterns of seasonal dynamics are very poorly reproduced either in timing or by the levels, or both (Figs. 7-10). Never mind the plausible oxygen dynamics in the surface layer, where it is mainly driven by air-sea gas exchange. [...]**

> *no objection*

**[...] Moreover, the focusing of analysis at the surface layer is unwarranted because the nitrogen biogeochemical cycle must be evaluated for the entire ecosystem, including sediments.**

> The sediment is represented by one layer in our model setup. The processes in the sediment and at the sediment-water interface are considerably simplified (see item (A) above). Hence, the sediment representation in the model is far to simple than one should take the modeled sediment concentrations for granted. Therefore, we prefer to evaluated the surface layer concentrations only (top $12$ m).

---

## Author Comment (AC4) · 12 Dec 2018

**Response to review comment #4 by referee #1**

We thank the reviewer for the feedback to this manuscript.

Below, the reviewer's comments are printed in bold letters and our answers in non-bold letters.

**1. Bioavailable PON has been mentioned at several places (such as line 21, page 1). This is a misleading term unless further qualified. Only inorganic nitrogen (dissolved nutrients form) and up to certain extent, DON is considered bioavailable. I have not seen any (oceanographic) study, where PON is proposed to be bioavailable. I am unsure if the authors wanted to convey the availability of nitrogen to heterotrophs (such as fish), then PON can be bioavailable. But in traditional view, we do not present the definition of bioavailable in this way.**

> We did not properly introduce the meaning of PON as we use it in the study. We denoted the sum of phytoplankton, zooplankton and detritus that is further available for the model processes as bioavailable PON. We will include this description.

**2. Authors must specify how does their model is able to differentiate between different components of anthropogenic (for that matter natural as well) inputs?**

> An established method to tag individual nutrient sources was used to trace *total nitrogen deposition*, *shipping-related nitrogen deposition*, and *agricultural-related nitrogen deposition* (p.7, l.14–16). The methods has originally been published by Ménesguen et al. (2006).

> When we write about *atmospheric nitrogen deposition* (without prefix) and *total atmospheric nitrogen deposition* we mean the sum of natural and anthropogenic nitrogen deposition. Hence, we do not differentiate between anthropogenic and natural inputs at all in this context.

> The shipping-related nitrogen deposition was calculated by the difference between atmospheric chemistry transport simulations with and without shipping emissions (p.5, l.28–29). We are aware that this approach is not the first choice for our use case – tagging shipping-nitrogen in the atmosphere would have been more appropriate – because non-linear interactions between shipping imissions and other atmospheric com-

pounds are neglected (*imissions* = atmospheric compounds that were *emitted* previously).

> The agricultural-related nitrogen deposition was estimated to be $95\,\%$ of reduced nitrogen deposition (only ammonia and ammonium; p.5, l.30–32). This assumption is roughly valid because $95\,\%$ to $100\,\%$ of the ammonia emissions originate from agricultural activities and animal husbandry. We are aware that we ignore agricultural-related oxidized nitrogen emissions into the atmosphere, e.g.: nitrate from wind blown dust, nitrogen oxide and dinitrogen oxide emissions from plants, and nitrogen oxide emissions from combustion-driven agricultural vehicles. Therefore, we state that our study underestimates the agricultural contribution (p.5, l.32–33).

**3. No uncertainties are provided in the estimates. It is important to provide uncertainties (in all tables and texts, wherever an estimate is quoted) anyways but here it more important as % contribution difference of different processes in the two basins is not much.**

> We agree with the reviewer that it is important to provide uncertainties. Unfortunately, we simulated only one year with one model and are not able to provide the variation between different years or within a model ensemble. Hence, we are not able to provide uncertainty estimates. We see no possibilty to sufficiently reply to this criticism.

> Due to the high (unknown) uncertainty we do no statistical evaluation of the model results.

**4. What deposition velocities are used in the model to estimate deposition rates? These must also have large uncertainties.**

> We are not sure if understood the question correctly. If we miss-understood the

question and answered it not correct, we kindly ask the reviewer to reformulate the
question.

> The manuscript is quite long, which was criticized by several other reviewers. We
were aware of that fact prior to submission and tried to keep some parts of the material
and methods section as short as possible (it is still very long). Hence, we provided
only brief references to publications, which explain the parameterization of the wet and
dry deposition processes in CMAQ (p.5, l.12–14). We wrote a bit more about it in
companion paper part A of this study (Neumann et al., 2018b; p.4, l.33 to p.6, l.2).

> A general overview of the processes and uncertainties: The wet deposition velocities
(and the scavanging rates) are substance specific and they are subject to uncertainty.
The nitrogen wet deposition considerably depends on the used meteorological forcing
(see Neumann et al. (2018a) and Karl et al. (2018, previously *in prep.*) for details).
In the case of particulate matter, the dry and wet deposition depends on the indiviual
particle size distribution at each time step and location. The dry deposition depends
on the surface roughness, which is not necessarily correctly represented in the model.
There is no bi-directional flux of gas phase species between atmosphere and ocean or
atmosphere and land surface included.

> We assessed the uncertainty introduced by the nitrogen deposition by using three
atmospheric nitrogen deposition data sets as forcing for HBM-ERGOM simulations in
anouther study (Neumann et al., 2018a). The differences between EMEP and our
CMAQ data were very high. Moreover, the resultion of the meteorological forcing also
impacted the nitrogen deposition pattern. Vivanco et al. (2017) evaluated the nitrogen
deposition of several atmospheric chemistry transport models and found a wide spread
in the results. In summary, yes, there are huge uncertainties in the atmospheric depo-
sition. However, we are not aware of any marine biogeochemical study, which properly
deals with the uncertainty in its atmospheric nutrient forcing.

**5. Chlorophyll is a pigment so how does one estimate relative contribution of**

**shipping etc to chlorophyll and what does it signify (Fig. 6)? Perhaps an estimate to primary production instead of chlorophyll would have been meaningful.**

> We did not describe the usage of chloropyll properly. Chlorophyll is calculated as diagnostic variable from the phyoplankton concentration. The relative contribution of nitrogen source XY to chlorophyll actually equals the quotient of *nitrogen of source XY in phytoplankton* divided by *total nitrogen in phytoplankton*. We have choosen chlorophyll for the evaluation because we validated chlorophyll and did not want to introduce another parameter. Considering primary production would have introduced additional uncertainty. We will include this description.

**6. Why a particular year (2012) is chosen (line 5, page 20)? Will the conclusions change for another year? How does one specify a particular year in model (unless there is some time-series analysis involved, which is not the case here)?**

> We used nitrogen deposition data that was obtained by atmospheric chemistry transport model simulations performed within the EU Bonus project SHEBA. In SHEBA, only the year 2012 was modelled because calculating high resolution emission data sets and removing artifacts from these data sets is very time consuming. We have choosen this nitrogen deposition data set and not e.g. EMEP data because the shipping contribution to the nitrogen deposition was provided by SHEBA. If we would have choosen EMEP data, we would have been able to simulate longer time periods. However, we would not have be able to properly quantify the shipping contribution. We are aware of the fact that considering only one year introduces a considerable bias.

> We plan to perform a decadal simulation with a more recent ERGOM version (coupled to MOM) and with EMEP deposition in future in order to deal with the question asked by the reviewer.

**7. Baltic Sea is also zone of nitrogen inputs through N2 fixation. Is this component taken into account in the model?**

> Yes, nitrogen fixation is included in the model. We forgot to mention it in the Material and Methods section. We will include it in the manuscript.

―――――――――――――――――